# FOCUS: HIGH-DIMENSIONAL MODEL-BASED RL VIA FOCUSED CONTROL UNIT SAMPLING

## ABSTRACT

Reinforcement learning (RL) in high-dimensional continuous state and action spaces often struggles with low learning efficiency and limited exploration scalability. To address this, we introduce FOCUS, a novel model-based RL framework that leverages the insight that effective policies often rely on dynamically focused, sparse control. FOCUS learns preferences over action dimensions to facilitate more targeted and efficient policy learning. It employs a hierarchical decision-making strategy, in which a high-level policy generates binary prompts to activate control units that have more impact on the task performance, while a low-level policy produces actions conditioned on these prompts. To promote behavioral diversity guided by different control-unit preferences, we integrate a diversity-driven objective into the model-based policy optimization process. FOCUS significantly outperforms existing methods on multiple visual control tasks. ~~Furthermore, it facilitates the integration of prior knowledge about the importance of action dimensions, making it particularly effective for complex, high-dimensional tasks.~~

## 1 INTRODUCTION

Model-based reinforcement learning (MBRL) provides a promising framework for decision-making tasks with high-dimensional observations by modeling environment dynamics (Hafner et al., 2020; 2025). However, its scalability is limited in high-dimensional action spaces, where exploration and policy optimization tend to become inefficient and unstable. In many real-world domains, such as humanoid locomotion, power grid control, and multi-agent collaboration, although the action spaces are high-dimensional, only a sparse subset of dimensions may be crucial for effective decision-making at each step. This insight highlights the need for mechanisms that can selectively attend to key control dimensions during learning, enabling more efficient and focused policy optimization.

A natural approach to mitigating the challenges of high-dimensional action spaces is to exploit their underlying structure. In particular, biasing policy learning toward more informative action dimensions can substantially reduce search complexity and improve efficiency. Several prior methods have explored this direction by introducing hierarchical or factorized action representations. However, many of these approaches rely on *discrete action assumptions* (Saito et al., 2024; Chen et al., 2019; Kumar et al., 2017) or *predefined base-action set* into which the original high-dimensional actions are decomposed (Kim & Dean, 2002; Geißer et al., 2020; Pierrot et al., 2021), thereby limiting their scalability to general continuous control tasks.

To address the under-explored problem of *jointly handling high-dimensional visual observations and high-dimensional continuous action spaces*, we propose FOCUS, a scalable MBRL framework. This setting presents unique challenges, including increased sample complexity and the difficulty of learning effective policies in expansive state-action spaces. Unlike existing MBRL methods (Hafner et al., 2025; Hansen et al., 2023) that treat all action dimensions uniformly, as shown in Figure 1, FOCUS introduces a *control-unit preference learning* mechanism that automatically identifies promising action subspaces and adaptively steers behavior learning toward them, enabling more efficient and focused behavior optimization and exploration.

Specifically, FOCUS represents control-unit preferences using learnable Bernoulli distributions over all action dimensions. The agent presents a *hierarchical decision-making strategy*: (i) A high-level preference policy predicts sampling probabilities and generates binary preference codes (the "prefer-

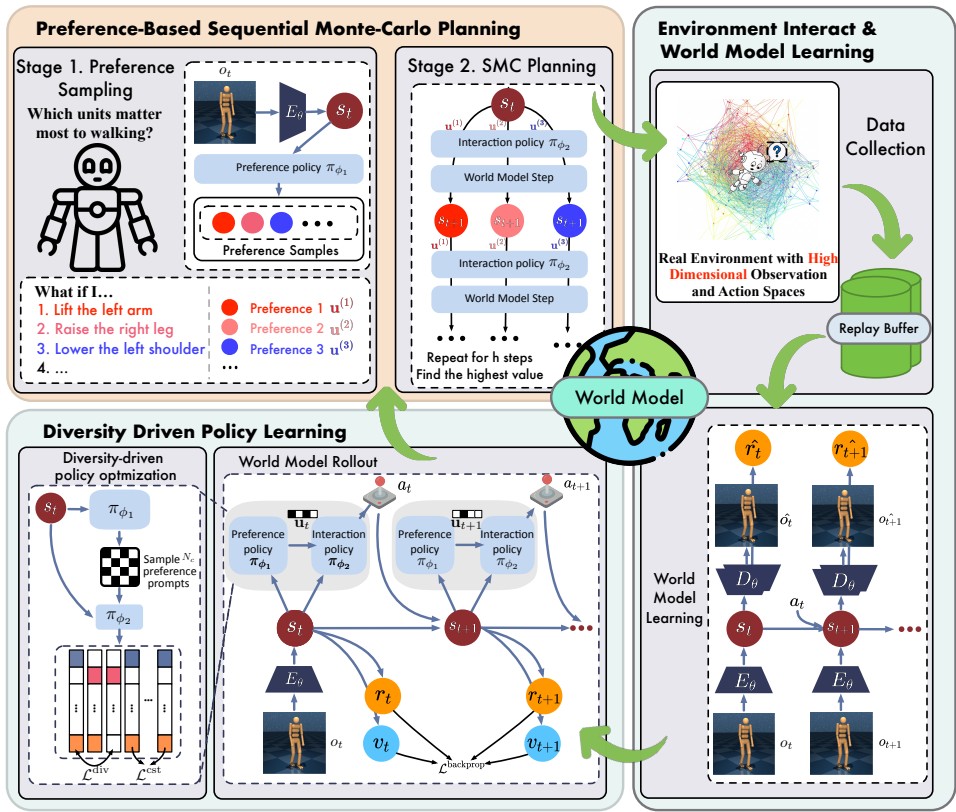

Figure 1: **FOCUS solves high-dimensional continuous control tasks.** The key idea is to improve policy optimization and exploration efficiency via a control-unit preference learning mechanism that adaptively selects and optimizes a dynamic subset of action dimensions.

ence prompt") over all control units; (ii) a low-level interaction policy, conditioned on this prompt, specializes in the selected action subspaces. Both policies are jointly optimized to maximize expected returns over imagined state rollouts produced by the world model. To encourage model-based exploration and alignment between decision layers, we introduce a *diversity-driven policy optimization objective* that maximizes behavioral discrepancies across different control-unit preferences. Once trained, the agent leverages the hierarchical policies for real-environment interaction by performing *preference-based Monte Carlo planning*, which reduces possible control-unit sampling bias and significantly improves the decision quality.

Experiments on benchmarks with high-dimensional action spaces, *i.e.*, DeepMind Control (Tassa et al., 2018), MyoSuite (Caggiano et al., 2022), and HumanoidBench (Sferrazza et al., 2024), demonstrate that FOCUS outperforms model-based and model-free RL approaches in both learning efficiency and final performance. In summary, the main contributions of this work are as follows:

- We present a unified framework for *high-dimensional MBRL*. It improves existing MBRL methods for efficient learning in large action spaces. Our method dynamically prioritizes promising control units for optimization and planning by modeling the importance of individual action dimensions.
- To enable the interaction policy to respond to varying control-unit preferences, we propose *diversity-driven model-based policy learning*, which leverages a world model to imagine and evaluate trajectories under diverse preference samplings.
- As an additional contribution, we introduce a *preference-based sequential Monte Carlo planning* algorithm, which delivers higher computational efficiency and decision quality than CEM based planning approaches in high-dimensional RL, including TD-MPC2 (Hansen et al., 2023).

## 2 PROBLEM FORMULATION

Our work targets high-dimensional continuous visual control tasks, where both the observation and action spaces are large and complex, posing significant challenges for efficient learning and

exploration. Formally, we formulate the problem as a Partially Observable Markov Decision Process (POMDP), defined by a 5-tuple $(\mathcal{O}, \mathcal{A}, \mathcal{R}, \mathcal{T}, \gamma)$, where: $\mathcal{O}$ is the high-dimensional visual observation space, $\mathcal{A}$ is the high-dimensional continuous action space, $\mathcal{R}$ is the reward function, $\mathcal{T}$ is the state transition dynamics, and $\gamma$ is the discount factor. In particular, we assume the action space is structured as a Cartesian product of $d$ continuous subspaces:

$$\mathcal{A} = \mathcal{A}^{(1)} \times \mathcal{A}^{(2)} \times ... \times \mathcal{A}^{(d)}, \quad \text{with } \mathcal{A}^{(i)} \subset \mathbb{R}. \tag{1}$$

An action at time step $t$ is thus represented as a $d$-dimensional vector $\left[a_t^{(1)}, a_t^{(2)}, \ldots, a_t^{(d)}\right]$, where each $a_t^{(i)} \in \mathcal{A}^{(i)}$ is referred to as the $i$-th sub-action. The objective is to learn a policy $\pi : \mathcal{O} \to \mathcal{A}$, that maximizes the expected discounted cumulative return $\mathbb{E}\left[\sum_{t=1}^{T} \gamma^{t-1} r_t\right]$.

**Challenges.** The entanglement of high-dimensional visual observations and high-dimensional continuous actions presents unique and under-explored challenges. Specifically, the state space induced by visual inputs (*e.g.*, raw images) is often noisy, redundant, and partially observable, requiring agents to infer latent dynamics over time. Meanwhile, large continuous action spaces lead to exponentially increasing possibilities for control decisions. Their intersection leads to an expansive and entangled state-action space, making exploration inefficient and value estimation difficult. As a result, existing RL methods often struggle with sample efficiency and generalization in such settings.

## 3 METHOD

### 3.1 OVERVIEW OF FOCUS

To tackle the challenges described above, we propose an MBRL method guided by learned control-unit preferences. The overall training and environment interaction pipeline consists of three stages:

  (i) *World model learning:* We first train a world model to predict latent state transitions and reward signals from observation-action-reward tuples. We use DreamerV3 (Hafner et al., 2025) as the backbone for world modeling. Detailed model specifications are provided in Appendix A

 (ii) *Diversity-driven model-based policy learning:* As shown in Algorithm 1, over future trajectory rollouts (*i.e.*, latent state imaginations) generated by the world model, FOCUS constructs a hierarchical policy learning architecture. At its core is a high-level decision-making module that outputs a preference distribution over all control units[1]. At each imagination step, the high-level policy samples a binary control-unit preference from Bernoulli distributions (termed a "*preference prompt*"). This prompt selects a sparse subset of control units, guiding the low-level policy network to generate specific behaviors within the original high-dimensional action space. The high-level *preference policy* and the low-level *interaction policy* are jointly optimized to maximize expected returns over the imagination horizon. Furthermore, we encourage behavioral diversity across different preference prompts, promoting more effective exploration and stronger alignment between hierarchical decisions.

(iii) *Preference-based Monte Carlo planning:* During environment interaction, the agent samples multiple candidate preference prompts. For each prompt, it performs multi-step Monte Carlo planning to generate potential future trajectories. The final action is then selected from these candidates based on trajectory weights computed from their cumulative advantage estimates.

### 3.2 CONTROL-UNIT PREFERENCE SAMPLING

Our approach is motivated by the observation that high-dimensional control tasks often exhibit inherent sparsity—at any given time, some control units more significantly contribute to task performance than others. For example, in the `HumanoidStand` task, the agent begins by pushing itself upward from the ground using support from its legs and arms. In such moments, joints such as the ankles and wrists play a dominant role, while many other action dimensions contribute minimally. This observation suggests that exploration can be made more efficient by selectively attending to a small subset of all control units. Accordingly, we introduce a high-level policy $\pi_{\phi_1}$ that takes latent states as inputs and generates a binary prompt $\mathbf{u}_t \in \mathbb{R}^d$, where $\mathbf{u}_t^{(i)} = 1$ indicates that the $i$-th action dimension

---

[1]Each *control unit* corresponds to a subset of action dimensions for a specific functional component (*e.g.*, a joint in a humanoid robot), potentially spanning multiple dimensions to reflect its degrees of freedom.

---

**Algorithm 1:** Full training algorithm of FOCUS, with Diversity-driven model-based policy learning, and Preference-based SMC Planning

---

1   **Initialize:** world model $\theta$, hierarchical actor $\{\phi_1, \phi_2\}$; critic $\psi$, replay buffer $\mathcal{B}$ with random episodes.
2   **while** not converged **do**
3     **for** update step $c = 1 \ldots C$ **do**
4       `// World model learning`
5       Draw data sequences $\{(o_t, a_t, r_t)\}_{t=1}^T \sim \mathcal{B}$.
6       Update the dynamics model $p_\theta$ and reward model $r_\theta$ using DreamerV3 world model objectives.
7       `// Diversity-driven model-based policy learning`
8       Draw a random observation $o_1 \sim \mathcal{B}$.
9       **for** time step $t = 1 \ldots H - 1$ **do**
10         Sample $N_c$ control-unit preference promts $\{\mathbf{u}_{t,n}\}_{n=1}^{N_c} \sim \pi_{\phi_1}(s_t)$.
11         Generate $\{a_{t,n} \sim \pi_{\phi_2}(s_t, \mathbf{u}_{t,n})\}_{n=1}^{N_c}$.
12         Predict $\{s_{t+1,n} \sim p_\theta(s_t, a_{t,n})\}_{n=1}^{N_c}$ and $\{r_{t,n} \sim r_\theta(s_{t+1,n})\}_{n=1}^{N_c}$.
13         Compute $\mathcal{L}_t^{\text{div}}$ and $\mathcal{L}_t^{\text{cst}}$ over $\{a_{t,n}\}_{n=1}^{N_c}$ according to Eq. equation 3 and Eq. equation 4.
14         Select $s_{t+1,n}$ with the highest $v_\psi(s_{t+1,n})$ as $s_{t+1}$.
15       **end**
16       Update $\pi_{\phi_1}, \pi_{\phi_2}$ and $v_\psi$ using Eq. equation 5.
17     **end**
18     `// Environment interaction with sequential Monte Carlo planning`
19     $o_1 \leftarrow$ `env.reset()`
20     **for** time step $t = 1 \ldots T$ **do**
21       Compute posterior state $s_t \sim q_\theta(o_t)$.
22       Sample $N_p$ preference promts $\{\mathbf{u}_n\}_{n=1}^{N_p} \sim \pi_{\phi_1}(\cdot \mid s_t)$.
23       Initialize $N_p$ planning trajectories $\{\tau_n\}_{n=1}^{N_p}$ starting from $\{\hat{s}_{t,n}\}_{n=1}^{N_p} = s_t$.
24       Initialize importance weights for planning trajectories $\{w_n = 1\}_{n=1}^{N_p}$.
25       **for** planning steps $h = 0 \ldots L - 1$ **do**
26         Generate $\{\hat{a}_{t+h,n} \sim \pi_{\phi_2}(\hat{s}_{t+h,n}, \mathbf{u}_n)\}_{n=1}^{N_p}$.
27         Predict $\{\hat{s}_{t+h+1,n} \sim p_\theta(\hat{s}_{t+h,n}, \hat{a}_{t+h,n})\}_{n=1}^{N_p}$ and $\{\hat{r}_{t+h,n} \sim p_\theta(\hat{s}_{t+h+1,n})\}_{n=1}^{N_p}$.
28         Compute step-wise advantages $\{A_{t+h,n} = \hat{r}_{t+h,n} + \gamma v_\psi(\hat{s}_{t+h+1,n}) - v_\psi(\hat{s}_{t+h,n})\}_{n=1}^{N_p}$.
29         Update weights $\{w_{t+h+1,n} = w_{t+h,n} \cdot \exp(A_{t+h,n})\}_{n=1}^{N_p}$.
30       **end**
31       Sample $\tau_n$ from a categorical distribution based on weights $\{w_{t+L,n}\}_{n=1}^{N_p}$.
32       Select the first action $\hat{a}_{t,n}$ in $\tau_n$ as $a_t$.
33       $r_t, o_{t+1} \leftarrow$ `env.step`$(a_t)$
34     **end**
35     Add experience to the replay buffer $\mathcal{B} \leftarrow \mathcal{B} \cup \{(o_t, a_t, r_t)\}_{t=1}^T$.
36   **end**

---

is currently important and should be prioritized. This design leads to a hierarchical policy structure, where the target policy $\pi_{\phi_2}$ is conditioned on both the latent state and the preference prompt:

$$\text{Preference policy:} \quad \mathbf{u}_t \sim \pi_{\phi_1}(s_t), \quad \text{Interaction policy:} \quad a_t \sim \pi_{\phi_2}(\mathbf{u}_t, s_t), \quad (2)$$

where $\mathbf{u}_t$ is sampled from $d$ independent Bernoulli distributions whose parameters are jointly predicted by the high-level policy $\pi_{\phi_1}$. Despite the independent sampling process, co-optimizing the high-level and low-level policies implicitly facilitates the discovery of interdependencies among action dimensions. For comparisons between independent sampled Bernoulli variables and explicitly enforced inter-unit dependencies, please refer to Appendix C.7. Intuitively, we expect the target policy $\pi_{\phi_2}$ to adapt its behavior based on the high-level guidance $\mathbf{u}_t$, enabling more focused and effective exploration within the subspace of prioritized action dimensions.

### 3.3   DIVERSITY-DRIVEN MODEL-BASED POLICY LEARNING

To promote an efficient exploration of potential policies guided by diverse control-unit activations, we introduce a focused policy optimization approach within a model-based behavior learning framework, as shown in Figure 1. We first learn a world model parameterized by $\theta$, which includes an encoder $p_\theta(s_t \mid o_t)$, a latent transition model $p_\theta(s_{t+1} \mid s_t, a_t)$, and a reward predictor $r_\theta(s_t)$. Given an initial

latent state encoded from a random observation in the replay buffer, FOCUS samples a preference prompt $\mathbf{u}_t$, and generates an preference-conditioned action $a_t$ based on the current rollout state $\hat{s}_t$. The action $a_t$ is then used to update the imagination trajectory via the world model. The hierarchical policy $\pi_{\phi_{1,2}}$ and the value model $v_\psi$ are optimized over imagined trajectories of length $H$.

However, directly maximizing the expected value function, as in Dreamer-style objectives, is insufficient for inducing low-level behaviors that are responsive to high-level preferences. Without further regularization, the actor may fail to interpret and follow the high-level guidance effectively. To address this, we design a diversity-driven learning objective. The key idea is that the low-level actor should produce diverse behaviors for control units selected by different preference prompts, while maintaining consistent behaviors for those not selected. Specifically, we sample $N_c$ preference prompts $\{\mathbf{u}_{t,n}\}_{n=1}^{N_c}$ from $\pi_{\phi_1}(s_t)$, and and generate corresponding action vectors $\{a_{t,n}\}_{n=1}^{N_c}$. Each action is decomposed as $a_{t,n} = [a_{t,n}^{(1)}, \ldots, a_{t,n}^{(d)}]$ along $d$ dimensions. We first define a diversity-promoting loss to encourage the low-level policy to respond distinctly to different preference selections:

$$\mathcal{L}_t^{\text{div}} = \sum_{i=1}^{d} \left[ -\sum_{n=1}^{N_c} \mathbb{I}(\mathbf{u}_n^{(i)} = 1) \sum_{k=1}^{N_c} \mathbb{I}(\mathbf{u}_k^{(i)} = 0) \cdot \text{KL} \left[ \pi(a_n^{(i)} \mid s, \mathbf{u}_n) \parallel \text{sg}(\pi(a_k^{(i)} \mid s, \mathbf{u}_k)) \right] \right], \quad (3)$$

where $\mathbb{I}(\cdot)$ is the indicator function, $\text{sg}(\cdot)$ denotes stop-gradient, and we ignore the time step index for clarity. Meanwhile, we introduce a consistency loss to penalize divergence among actions on the same dimensions not selected by the preference prompt:

$$\mathcal{L}_t^{\text{cst}} = \sum_{j=1}^{d} \left[ \sum_{n=1}^{N_c} \mathbb{I}(\mathbf{u}_n^{(j)} = 0) \sum_{k=1}^{N_c} \mathbb{I}(\mathbf{u}_k^{(j)} = 0) \cdot \text{KL} \left[ p(a_n^{(j)} \mid s, \mathbf{u}_n) \parallel \text{sg}(p(a_k^{(j)} \mid s, \mathbf{u}_k)) \right] \right]. \quad (4)$$

Together, these losses impose a contrastive structure in the high-dimensional action space, enhancing exploration and ensuring better alignment between low- and high-level policies. Importantly, we do not stop the gradient at $\mathbf{u}_n$ in the above regularization terms. Instead, we allow $\pi_{\phi_1}$ to be co-optimized with $\pi_{\phi_2}$. This design further promotes exploration, as $\mathcal{L}_t^{\text{div}}$ rewards diverse control-unit activations and prevents the high-level policy from collapsing to identical but biased activation vectors.

For the sake of computational efficiency, as shown in Alogrithm 1, we choose $s_{t+1,n} \sim p_\theta(s_t, a_{t,n})$ with the highest predicted value $v_\psi(s_{t+1,n})$ as $s_{t+1}$ among multiple candidates at each imagination step. In practice, we compute the diversity-driven objectives only at the first step of imagination. Despite this simplification, we observe a significant performance gain over versions without diversity-driven regularization, while incurring almost no additional training overhead.

Finally, the full objective is defined as follows, where the actor uses value backpropagation through dynamics as in DreamerV3 for better continuous control, and the critic uses a $\lambda$-return target:

$$\texttt{Actor:} \quad \mathcal{L}(\phi_1, \phi_2) = \mathbb{E}_{p_\psi, p_\theta} \left[ \sum_{t=1}^{H-1} \left[ \mathcal{L}_t^{\text{div}} + \mathcal{L}_t^{\text{cst}} - V_t^\lambda - \eta \mathcal{H}\left(a_t \mid \hat{s}_t, \text{sg}(\mathbf{u}_t)\right) \right] \right],$$

$$\texttt{Critic:} \quad \mathcal{L}(\psi) = \mathbb{E}_{p_{\phi_1}, p_{\phi_2}, p_\theta} \left[ \sum_{t=1}^{H-1} \frac{1}{2} (v_\psi(\hat{s}_t) - \text{sg}(V_t^\lambda))^2 \right], \quad (5)$$

where the entropy term $\mathcal{H}(\cdot)$ further encourages exploration conditioned on preference prompts. Notably, both levels of policies are jointly optimized towards maximizing the imagined values.

### 3.4 Preference-Based Monte Carlo Planning

The hierarchical policy involves two sources of stochasticity—from both the high-level preference sampling and the low-level action generation—which may increase variance during real-environment interaction. To mitigate this, we develop an online planning scheme that efficiently reasons over multiple future state-action trajectories, each conditioned on a different control-unit preference prompt. We term this method *preference-based Monte Carlo planning*, a modified form of sequential Monte Carlo planning (Piché et al., 2018) designed for high-dimensional action spaces.

As shown in Algorithm 1, at each interaction step, we sample $N_p$ preference prompts $\{\mathbf{u}_n\}_{n=1}^{N_p}$ based on the current state $s_t$, and initialize $N_p$ planning trajectories with uniform weights $w_n = 1$. At

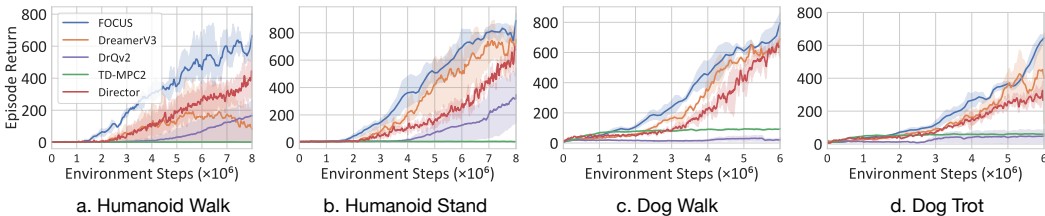

**Figure 2: Results on DeepMind Control Suite.** All results are averaged over three training seeds.

each planning step $t + h$, we reuse $\mathbf{u}_n$ to sample low-level actions $\hat{a}_{t+h,n} \sim \pi_{\phi_2}(\hat{s}_{t+h,n}, \mathbf{u}_n)$. This contrasts with model-based policy learning, where $\mathbf{u}_n$ is typically re-sampled at each imagination step. Instead, we preserve temporal consistency in the sampled control-unit prompts during planning. We then use the learned dynamics $p_\theta$ to rollout future states $s_{t+h:t+L,n}$. We evaluate these candidate trajectories via advantage estimation and update their weights accordingly:

$$A_{t+h,n} = \hat{r}_{t+h,n} + \gamma v_\psi(\hat{s}_{t+h+1,n}) - v_\psi(\hat{s}_{t+h,n}), \quad w_{t+h+1,n} = w_{t+h,n} \cdot \exp(A_{t+h,n}). \quad (6)$$

At the end of the planning horizon, we sample one trajectory from the categorical distribution defined by $\{w_{t+L,n}\}_{n=1}^{N_p}$ and execute its first action in the environment. Empirically, this approach significantly outperforms prior sample-based planning methods (*e.g.*, MPPI in TD-MPC2 (Hansen et al., 2023)) in both computational efficiency and decision quality. Unlike MPPI, our method avoids online policy refitting, making it more suitable for high-dimensional control tasks.

## 4 EXPERIMENTS

### 4.1 EXPERIMENTAL SETUP

First, we conduct experiments on two RL environments with visual inputs, namely DeepMind Control Suite (Tassa et al., 2018) and MyoSuite (Caggiano et al., 2022). We present empirical results on seven tasks from these environments with high-dimensional continuous action spaces. To demonstrate the applicability of our diversity-driven hierarchical policy learning, we conduct additional experiments on two tasks of HumanoidBench (Sferrazza et al., 2024) with proprioceptive robot state as input. Detailed information on the selected tasks with their action space sizes is provided in Appendix B.

We compare FOCUS with: (1) *DreamerV3* (Hafner et al., 2025), a strong model-based RL method in high-dimensional observation space; (2) *TD-MPC2* (Hansen et al., 2023), a decoder-free model-based RL baseline that employs the CEM-based planning method; (3) *DrQv2* (Yarats et al., 2021a), a data augmentation-based visual RL approach; (4) *Director* (Hafner et al., 2022), another hierarchical MBRL method that conditions low-level policy on goals generated by the high-level autoencoder. Particularly for HumanoidBench, the previous model-based methods suffer from computational overhead. While FastTD3 (Seo et al., 2025) has demonstrated superior efficiency in solving HumanoidBench tasks, we make modifications to this model-free baseline to create a new variant named FastTD3-FOCUS, and compare it to the vanilla FastTD3. The modified version substitutes hierarchical policy for a flat one, incorporating diversity-driven learning objective in FOCUS, without learning a world model and execute SMC planning based on the learnt model.

All models are trained under a fixed environment interaction budget. For two tasks in the `Humanoid` suite, each model is trained for 8M frames, and for the `Dog` suite, 6M frames. Tasks in `MyoSuite` allow interactions for 1M frames, and 500K for tasks in `HumanoidBench`. Average episode returns are computed over 10 episodes per seed.

### 4.2 DOMAIN #1: DEEPMIND CONTROL SUITE

As shown in Figure 2, FOCUS consistently outperforms baseline methods across all four tasks. Notably, it achieves a substantial improvement over DreamerV3 in `HumanoidWalk`. In both `Humanoid` tasks, the learning curves show a prolonged warm-up phase before meaningful improvement, reflecting the challenge of locating sparse high-reward regions within large decision spaces. Compared with **DreamerV3**, our model exhibits an earlier performance rise, demonstrating its ability to reduce sample complexity and efficiently explore targeted reward regions. In `Dog` tasks, rewards are less sparse than in `Humanoid` tasks, and DreamerV3 is already capable of sampling high-reward regions. This is reflected in its steadily increasing learning curves, suggesting that despite the larger

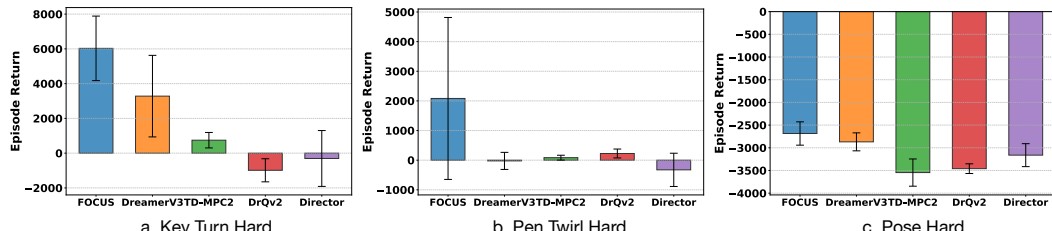

Figure 3: **Results on the MyoSuite hard tasks.** Episode returns are averaged over 10 episodes per seed, for a total of 30 runs. FOCUS outperforms all baselines across every task.

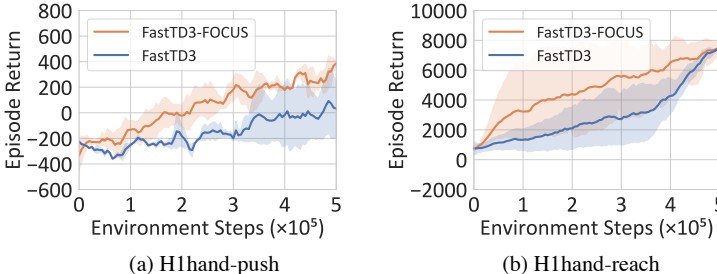

(a) H1hand-push        (b) H1hand-reach

Figure 4: **Results on HumanoidBench.** We implement FOCUS on top of FastTD3 rather than DreamerV3 or TD-MPC2, as these models fail to converge within the same number of interaction steps, likely due to limited interaction efficiency. FastTD3, in contrast, is specifically optimized for efficient environment interaction. All results are averaged over three training seeds.

number of action dimensions, the high-reward regions in the action space are sufficiently large for agents with random policies to sample with reasonable probability. In this setting, exploration is not the primary bottleneck, which limits potential performance gains. For **TD-MPC2**, it fails to achieve meaningful progress on `humanoid` tasks, highlighting its difficulty in effectively exploring the vast search space, particularly under high-dimensional visual input. The comparison with **Director** highlights the impact of the diversity-driven objective: while both methods employ hierarchical policies, FOCUS explicitly prioritizes subsets of the action space, resulting in higher exploration efficiency. We showcase the policy evaluation of FOCUS, DreamerV3 and Director in **??**, with full results in Appendix C.1. As illustrated, the policy obtained by FOCUS adjusts humanoid's posture in fewer time steps to start walking ($t = 50$), meanwhile maintaining its control more stable than DreamerV3 which falls to the ground ($t = 200$) after standing up. Additionally, DrQv2 requires roughly 4M environment steps before its learning curve begins to rise, suggesting that augmenting failing episodes early has little effect on sample efficiency.

We provide further results, comparing the planning efficiency of FOCUS with TD-MPC2 in Appendix C.2, and evaluating its generalization to low-dimensional action spaces in Appendix C.4.

### 4.3 DOMAIN #2: MYOSUITE

Performance results on `MyoSuite` are shown in Figure 3. FOCUS outperforms all baselines on `KeyTurnHard` and `PenTwirlHard`. On `KeyTurnHard`, it improves over DreamerV3, demonstrating higher efficiency in exploring high-reward regions within the same environment step budget On `PenTwirlHard`, FOCUS successfully discovers sharply peaked reward regions, whereas other baselines struggle in flatter areas; DreamerV3 fails to solve this task. The integration of the diversity-promoting loss $\mathcal{L}_t^{div}$ and consistency loss $\mathcal{L}_t^{cst}$ systematically biases sampling toward high-reward regions, enabling efficient policy learning and sustained performance gains All methods face challenges on `PoseHard`, though FOCUS achieves slightly better results, thanks in part to $\mathcal{L}_t^{cst}$ preventing overfitting as discussed in Section 3.3.

### 4.4 DOMAIN #3: HUMANOIDBENCH

We present the results of two tasks on `HumanoidBench` in Figure 4, while results of extensive 6 tasks could be found in Figure 16. We implement FOCUS on top of FastTD3 rather than DreamerV3 or TD-MPC2, as these models fail to converge within the same number of interaction steps, likely due to limited interaction efficiency. FastTD3, in contrast, is specifically optimized for efficient

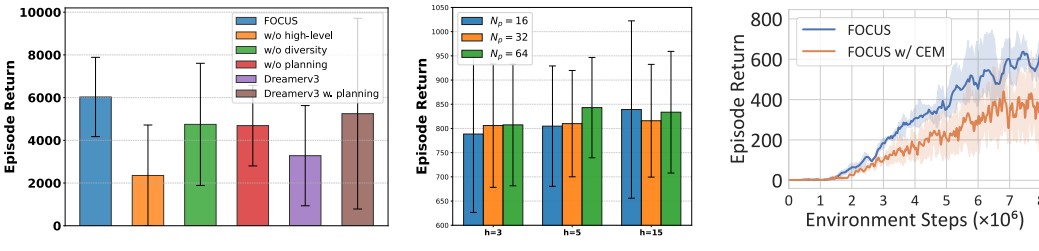

(a)                          (b)                          (c)

Figure 5: **Ablation studies.** (a): Effect of each model component (`KeyTurnHard`). (b): Effect of planning horizon $h$ and candidate trajectories $N_p$ in our planner (`HumanoidWalk`). (c): Ablation study comparing CEM and SMC planning (`HumanoidWalk`).

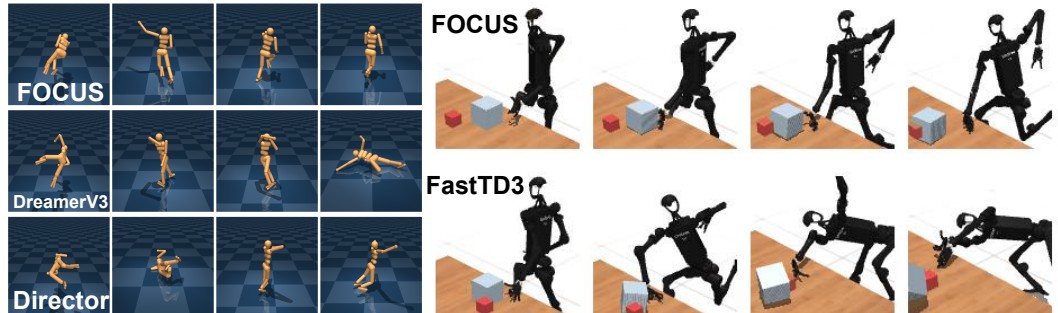

Figure 6: **Visualization of the learned policies.** We compare different models on `HumanoidWalk` (Left) and `H1hand-push` from `HumanoidBench` (Right). In `H1hand-push`, the robot's task is to push the object to the red target location.

environment interaction. Details of FastTD3 and our modifications are provided in Appendix C.5. In both `H1hand-push` and `H1hand-reach`, FastTD3-FOCUS achieves better sample efficiency than the vanilla FastTD3, which uses a simple MLP policy. As shown in Figure 4a for `H1hand-push`, FastTD3-FOCUS maintains consistent improvement throughout training, indicating that focused exploration helps the agent avoid local optima where the flat MLP policy is prone to getting stuck. Similarly, in the `H1hand-reach` task (Figure 4b), although both methods eventually converge to similar episodic returns, FastTD3-FOCUS demonstrates higher sample efficiency. This follows from the diversity-driven objective, which encourages distinct and meaningful behavior patterns among control units, enabling faster coverage of high-return regions of the state space and accelerating the acquisition of effective reaching behaviors.

## 4.5 MODEL ANALYSES

**Ablation studies.** We validate the proposed modules on `KeyTurnHard` in MyoSuite, considering three variants of our method: (1) FOCUS without $\pi_{\phi_1}$, ~~reducing it to a DreamerV3 agent planning in the original high-dimensional action space~~; (2) FOCUS without $\mathcal{L}_t^{div}$ and $\mathcal{L}_t^{cst}$; and (3) FOCUS without planning. These variants are compared with the full FOCUS, DreamerV3, and DreamerV3 w. planning to assess the contribution of each component. As shown in Figure 5(a), FOCUS consistently outperforms all other variants, demonstrating that each component contributes to its success. The variant without the high-level policy $\pi_{\phi_1}$ is equivalent to DreamerV3 with the addition of a Sequential Monte Carlo planner during both exploration and policy execution, yet it performs worse than DreamerV3 on account of biased exploration under noised value guidance. DreamerV3 with planning shows improved but more unstable behavior, indicating vulnerability to being trapped in local optima. Instead of planning in the flat action space, FOCUS plans in high level which promotes diversity, effectively bypassing these optimization traps during training. Furthermore, both the diversity loss and the planning algorithm are crucial contributors to FOCUS's performance. We also compare FOCUS to its Cross Entropy Method (CEM)-based variant in Figure 5(c). As illustrated, SMC enables faster and more effective learning compared to CEM. We argue that prior sample-based planning methods like CEM, which require online distribution refitting, are not sufficiently capable of modeling target distributions in a high-dimensional action space and thus limit their effectiveness.

**Hyperparameter analyses.** We examine different hyperparameter combinations for planning horizons $h$ and the number of candidate trajectories $N_p$ in Figure 5 (b) and **??**. We find that increasing

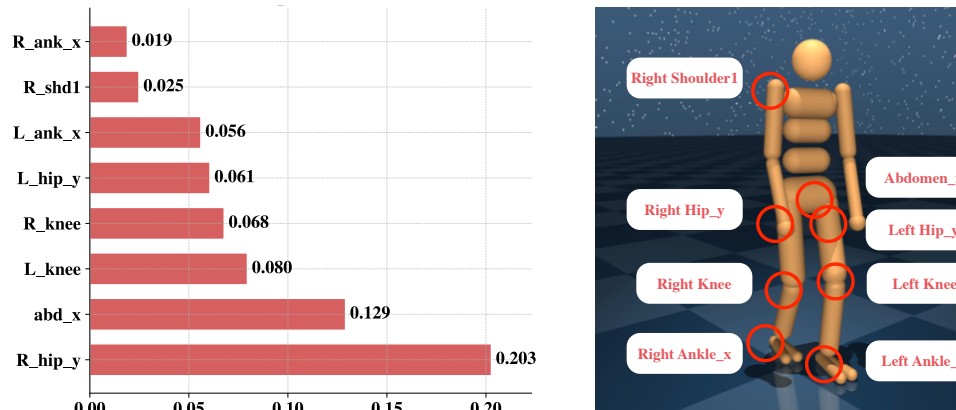

Figure 7: **Visualization of most frequently selected action dimensions.** The sparse action dimensions highlighted by FOCUS align closely with the kinematic priorities of the `HumanoidStand` task. demonstrating that the model effectively identifies behaviorally relevant control channels.

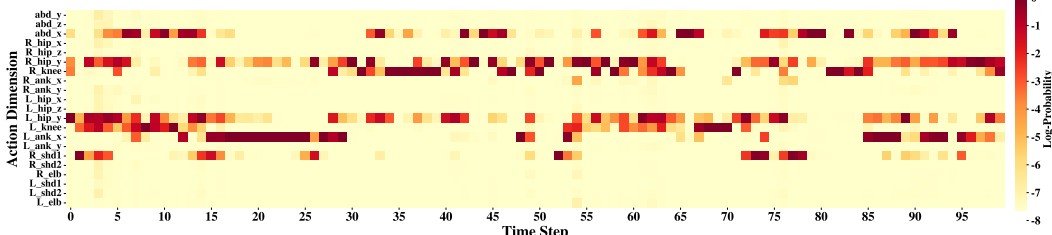

Figure 8: **Temporal evolution of action preferences in Bernoulli distributions over the first** $100$ **timesteps.** The high-level preference policy employs a sparse and dynamic selection strategy, with specific action dimensions being selectively activated in response to temporal transitions.

$N_p$ generally improves overall performance, but does not guarantee a monotonic increase in episode return. We attribute this to the exclusive use of the policy log-likelihood in the advantage calculation, which can make planning riskier. Including the log-likelihood term is standard practice in previous Control-as-Inference research (Levine, 2018; Piché et al., 2018), but in high-dimensional action spaces it introduces large variance, leading to rapid particle degeneration. Removing this term mitigates variance but promotes riskier planning, as actions with larger magnitudes are weighted higher without accounting for the probability of being sampled.

### 4.6 VISUALIZATION OF LEARNED CONTROL-UNIT PREFERENCES

We further present quantitative and qualitative results regarding the preference prompts sampled on `HumanoidStand`, using the model trained immediately after the rising point (*i.e.*, after 3M environment steps) and recording preference prompts over a single episode. These results serve two purposes: (1) to qualitatively assess whether the learned preference policy captures meaningful, structured sparsity across action dimensions, and (2) to illustrate the temporal evolution of control preferences as the agent attempts to stand.

We visualize the 8 most frequently selected action dimensions on the humanoid embodiment along with their selection frequencies in Figure 7. The full action dimension frequencies are presented in Appendix C.8, demonstrating that FOCUS successfully captures the structural sparsity within the original action space. Actuators corresponding to these preferred dimensions are highlighted to showcase FOCUS's ability to discover meaningful kinematic correspondences. As illustrated, the learned preferences are strongly concentrated on the lower-body actuators and the core trunk, which are physically critical for the `HumanoidStand` task. Specifically, the hip joints (*e.g.*, *R_hip_y*) and abdomen (*abd_x*) dominate the selection, reflecting their key role in generating upward torque and maintaining core stability necessary for an upright posture. This distribution confirms that FOCUS effectively identifies structural sparsity, facilitating efficient policy learning in a reduced action space.

In Figure 8, we further visualize the temporal evolution of the learned preference prompts in Bernoulli distributions. The results show that rather than converging to a static set of actuators, the policy dynamically modulates its focus across different dimensions over time, adapting to time-varying task

requirements. Together, these results support the hypothesis that FOCUS leverages structured sparsity to concentrate exploration on behaviorally relevant subspaces, aligning with our goal of reducing exploration overhead in high-dimensional control tasks.

## 5 RELATED WORK

**Model-based RL.** Model-based RL (Ha & Schmidhuber, 2018; Hafner et al., 2025; Hansen et al., 2023; Moerland et al., 2023) shows great potential in addressing the data efficiency issue compared to model-free methods (Yarats et al., 2021b; Kostrikov et al., 2021; Laskin et al., 2020), particularly in online learning. By leveraging a learned world model, these methods enable efficient policy optimization through planning (Nguyen et al., 2021; Zhao et al., 2021; Hansen et al., 2023) or Q-learning (Hafner et al., 2020; Wang et al., 2022). However, in high-dimensional observation and action spaces, world models struggle to identify sparse high-reward regions due to their uniform treatment of all action dimensions. This limitation burdens the learning process, as undirected exploration in vast action spaces leads to inefficiency. FOCUS addresses this challenge by adaptively selecting a control unit, thereby guiding policy exploration and optimization within a reduced subspace.

**Hierarchical RL.** Prior studies in hierarchical structured action space (Kumar et al., 2017; Chen et al., 2019; Tang & Agrawal, 2020; Saito et al., 2024) usually assume a multi-stage policy that divides the searching space into smaller parts. However, this structure is mostly restricted to tasks with discrete action spaces and requires additional discretization when confronted with continuous ones. Hierarchical policy learning in behavior learning (Pateria et al., 2021) has been used to enable efficient long-horizon reasoning by generating temporal abstractions through high-level policy (Vezhnevets et al., 2017; Gumbsch et al., 2024; Gürtler et al., 2021). Director (Hafner et al., 2022) is a model-based method featuring a hierarchical policy structure that generates a goal with a high-level autoencoder first. Yet in our work, we have made use of the hierarchical structure to model the sparsity nature of the high-dimensional action space, thus reducing the searching space to a single control unit and guiding the exploration in a high-dimensional continuous action space.

**RL with high-dimensional action space.** Despite the hierarchical structure mentioned above, methods that exploit prior knowledge to reduce exploration can efficiently deal with high-dimensional action space. Factored action space (Osband & Van Roy, 2014; Guestrin et al., 2001; Mahajan et al., 2021; Peng et al., 2021) decompose the original MDP into several sub-questions. Other works focused on over-actuated system and leverage the prior of musculoskeletal systems to reduce action space (Schumacher et al., 2022; Luo et al., 2023). Leaning in high-dimensional action space becomes difficult when prior knowledge is inaccessible. Similar to FOCUS, it's intuitive to learn from data the redundancy within action space.(Chandak et al., 2019; Baram et al., 2021; Zhong et al., 2024), while these methods either suffer from generalization or be specific to discrete action spaces. Additionally, learning-based structure discovery methods (Tavakoli et al., 2017; 2018; Van de Wiele et al., 2020; Zhang & Cai) are trained to discover underlying structure with separate actors or deterministic action mask. However, these designs either ignore action dimensional interdependencies, or suffer from drastic-shifting dynamics, which can lead to instability during training.

## 6 CONCLUSIONS AND LIMITATIONS

In this paper, we introduced FOCUS, a novel approach for high-dimensional reinforcement learning that addresses the challenges of learning and exploration efficiency in large action spaces. FOCUS leverages a hierarchical policy structure, where a high-level policy learns to generate sparse control-unit activations, identifying key action dimensions at each decision point; while a low-level policy operates within these preferred subspaces to optimize behavior using imagined rollouts from a world model. Our empirical results showed that FOCUS consistently outperforms baselines on high-dimensional control tasks while maintaining high learning efficiency.

Despite its strong performance, FOCUS also has certain limitations. First, the effectiveness of the learned preference prompts is closely tied to the quality of the world model, as inaccuracies in imagined rollouts can lead to suboptimal high-level guidance. Second, the hierarchical decision-making mechanism introduces extra parameters and training complexity. Exploring effective preference representations can be a promising direction for future work.

## ETHICS STATEMENT

This work proposes an RL framework focusing on dealing with high-dimensional action spaces. The research does not involve human subjects, sensitive personal data, or any newly collected datasets. All experiments were conducted within simulated environments, ensuring that no privacy or security concerns were introduced.

Our study does not directly present harmful applications. The contributions are methodological and algorithmic in nature, aiming to deepen the understanding of efficient policy learning in high-dimensional spaces. We have carefully read and followed the Code of Ethics, and are committed to principles of research integrity. No conflicts of interest or external sponsorship influenced this work.

## REPRODUCIBILITY STATEMENT

This work proposes a novel model, FOCUS, to deal with RL in tasks with high-dimensional observation and action input. To facilitate reproducibility, we provide the complete implementation code as part of the supplementary materials. Detailed descriptions of the model is included in Section 3.2, Section 3.3 and the planning algorithm is included in Section 3.4. We also describe the detailed experimental settings in Section 4.1 and Appendix B.

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

APPENDIX

The technical appendix includes:

- Section A: A detailed description of the world model architecture and its learning objectives.
- Section B: An overview of the action space sizes for all visual control tasks used in our experiments.
- Section C: Additional quantitative and qualitative results, including:
    - Section C.1: Visualization of learned policies.
    - Section C.2: Comparison of the planning efficiency per environment interaction step with TD-MPC2, which is based on MPPI planning.
    - Section C.3: Visualizations of the learned action-dimensional preferences, demonstrating that the high-level policy generates focused and interpretable control-unit prompts.
    - Section C.4: Additional results of FOCUS on low-dimensional tasks, highlighting its versatility across diverse visual control settings.
    - Section C.5: Detailed experimental settings for `HumanoidBench`.
    - Section C.6: Hyperparameter sensitivity analyses on `DogWalk`.
    - Section C.7: Ablation studies on alternative high-level policy network designs that enforce explicit dependencies across action dimensions, rather than sampling them independently.
    - Section C.8: Full action-dimension selection frequencies, illustrating successful reduction of the effective action space.
    - Section C.9: Full empirical results on HumanoidBench with extensive 6 tasks, further demonstrating the efficiency of FOCUS's hierarchical diversity-driven policy learning in high-dimensional action space.
- Section D: The description of LLM usage in this work.

## A WORLD MODEL DETAILS

The DreamerV3 world model is implemented as a recurrent state-space model (RSSM), which maintains a deterministic hidden state $h_t$ and a stochastic latent variable $z_t$ to encode observations and predict future states. In our main text, we denote the concatenated feature $s_t = [h_t, z_t]$ as the overall state representation. The world model can be formulated as follows:

$$
\begin{aligned}
\text{Sequence model:} \quad & h_t = f_\phi(h_{t-1}, z_{t-1}, a_{t-1}), \\
\text{Encoder:} \quad & z_t \sim q_\phi(z_t \mid h_t, x_t), \\
\text{Dynamics predictor:} \quad & \hat{z}_t \sim p_\phi(\hat{z}_t \mid h_t), \\
\text{Reward predictor:} \quad & \hat{r}_t = r_\phi(h_t, z_t), \\
\text{Continue predictor:} \quad & \hat{c}_t = c_\phi(h_t, z_t), \\
\text{Decoder:} \quad & \hat{x}_t = g_\phi(h_t, z_t),
\end{aligned}
\tag{7}
$$

where $f_\phi$ is the recurrent transition function modeled by a GRU. $q_\phi$ and $p_\phi$ are implemented with MLPs. We use another MLP with separate heads $r_\phi$ and $c_\phi$ to predict the reward $r_t$ and the episode continuation flag $c_t$, respectively. We use a CNN decoder to reconstruct the observation $x_t$.

The world model is trained by minimizing a variational objective that combines reconstruction losses with KL regularization. In particular, the total loss is written as:

$$
\begin{aligned}
\mathcal{L}(\phi) &= \mathbb{E}_{\tau \sim \mathcal{B}} \left[ \sum_{t=1}^{T} \left( \mathcal{L}_t^{\text{pred}} + \mathcal{L}_t^{\text{dyn}} + \mathcal{L}_t^{\text{rep}} \right) \right], \\
\mathcal{L}_t^{\text{pred}} &= - \ln p_\phi(x_t \mid h_t, z_t) - \ln p_\phi(r_t \mid h_t, z_t) - \ln p_\phi(c_t \mid h_t, z_t), \\
\mathcal{L}_t^{\text{dyn}} &= \text{KL}\left( \text{sg}(q_\phi(z_t \mid h_t, x_t)) \,\|\, p_\phi(\hat{z}_t \mid h_t) \right), \\
\mathcal{L}_t^{\text{rep}} &= \text{KL}\left( p_\phi(\hat{z}_t \mid h_t) \,\|\, \text{sg}(q_\phi(z_t \mid h_t, x_t)) \right).
\end{aligned}
\tag{8}
$$

where $\mathcal{L}_t^{\text{pred}}$ is a self-supervised loss that reconstructs $x_t$ and predicts the reward $r_t$ and terminal signal $c_t$; $\mathcal{L}_t^{\text{dyn}}$ is a KL divergence aligning the prior $p_\phi(\hat{z}_t \mid h_t)$ to the posterior $q_\phi(z_t \mid h_t, x_t)$; and $\mathcal{L}_t^{\text{rep}}$ is a reverse KL that regularizes the latent representations.

Table 1: **Overview of action space dimensions for all continuous control tasks in our experiments.**

| Environment | Tasks | Action Space Size ($\mathbb{R}^d$) |
|---|---|---|
| High-dimensional DMC | Humanoid Stand | 21 |
| | Humanoid Walk | 21 |
| | Dog Walk | 38 |
| | Dog Trot | 38 |
| MyoSuite | Key Turn Hard | 39 |
| | Pen Twirl Hard | 39 |
| | Pose Hard | 39 |
| Low-dimensional DMC | Walker Walk | 6 |
| | Cheetah Run | 6 |
| | Quadruped Walk | 12 |

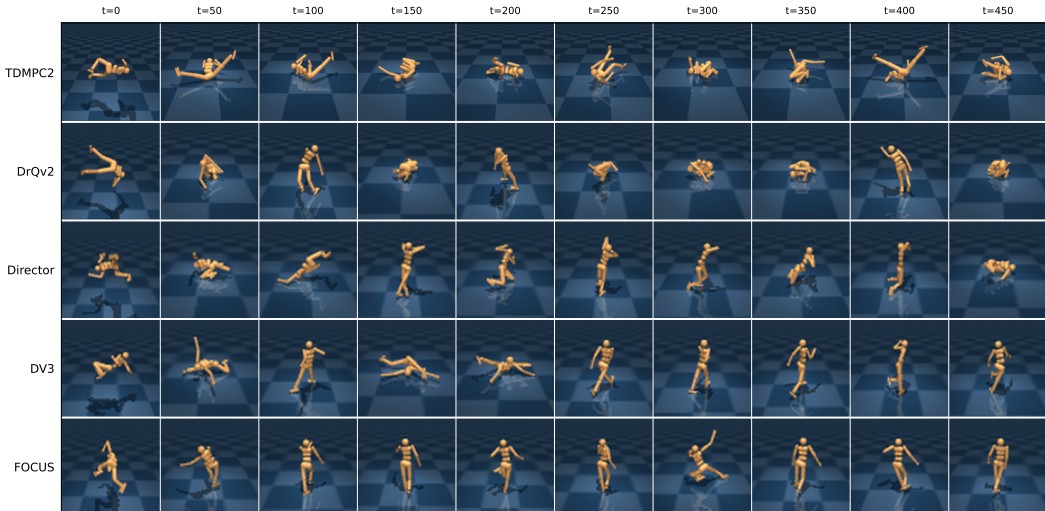

Figure 9: **Demonstration of the learned policy on `HumanoidWalk`.**

## B  SUMMARY OF EXPERIMENTAL ENVIRONMENTS

Table 1 summarizes the environmental setups in the DeepMind Control and MyoSuite benchmarks. While FOCUS is designed to tackle high-dimensional problems with large action spaces, its results on low-dimensional tasks highlight broad applicability.

## C  ADDITIONAL QUANTITATIVE AND QUALITATIVE RESULTS

### C.1  VISUAL EVALUATION OF LEARNT POLICIES

We evaluate trained policies of different models on the DMC and Myosuite tasks and visualize episode frames for comparison. The full results on `HumanoidWalk`, `DogTrot`, and `KeyTurnHard` are presented in Figure 9, Figure 10 and Figure 11 respectively.

### C.2  PLANNING EFFICIENCY COMPARISON WITH TD-MPC2

We compare the planning efficiency of FOCUS and TD-MPC2 in Figure 12, measuring the time consumed from receiving an observation to producing an action. The experiments were conducted on a system equipped with an NVIDIA RTX 4090 GPU and a dual-socket Intel Xeon Gold 6240C CPU at 2.60GHz. For a fair comparison, both compared models are implemented in PyTorch 2.4.1, whereas the proposed FOCUS is implemented in JAX for the main text and all other experiments.

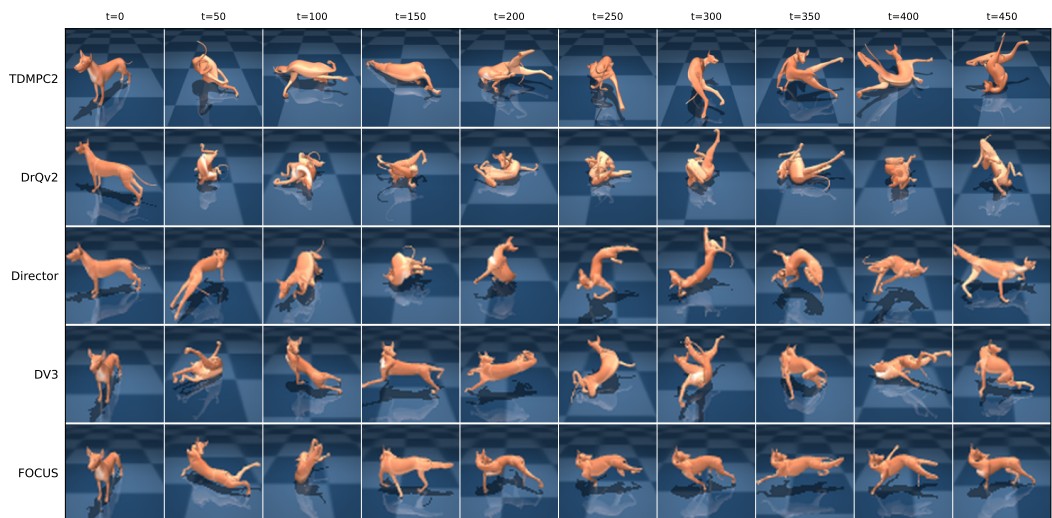

Figure 10: **Demonstration of the learned policy on `DogTrot`.**

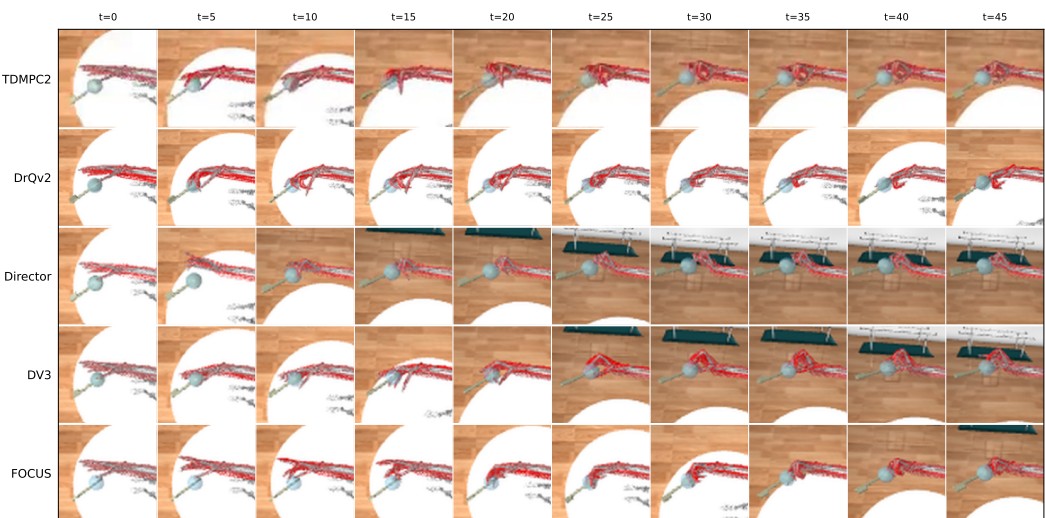

Figure 11: **Demonstration of the learned policy on `KeyTurnHard`.**

Although both methods rely on online planning, TD-MPC2 uses the *Model Predictive Path Integral* (MPPI) algorithm which is generally more computationally intensive than the proposed Sequential Monte Carlo (SMC) planning employed in FOCUS. As shown, FOCUS achieves significantly lower latency than TD-MPC2, with a notably smaller increase in latency across varying planning horizons. This also highlights the scalability of FOCUS for long-horizon tasks.

### C.3    VISUALIZATION OF LEARNED HIGH-LEVEL PREFERENCES

We visualize the preference prompts sampled by the learned high-level policy in Figure 13, and present the full visualization video in supplementary materials. The results show that our hierarchical policy can selectively prioritize action dimensions corresponding to key joints, such as the legs and arms in the `humanoid` tasks.

For instance, in the second row in Figure 13, the policy focuses on the *left knee* and the *x-actuator at the right ankle* to support the stand-up motion. A closer look at joint control axes reveals that the selected x-actuator at the right ankle is oriented vertically to the ground, making it particularly effective for initiating a standing-up behavior. In the third row, upon detecting that the right lower leg requires adjustment, the policy prioritizes control of the right knee joint to enable stable locomotion.

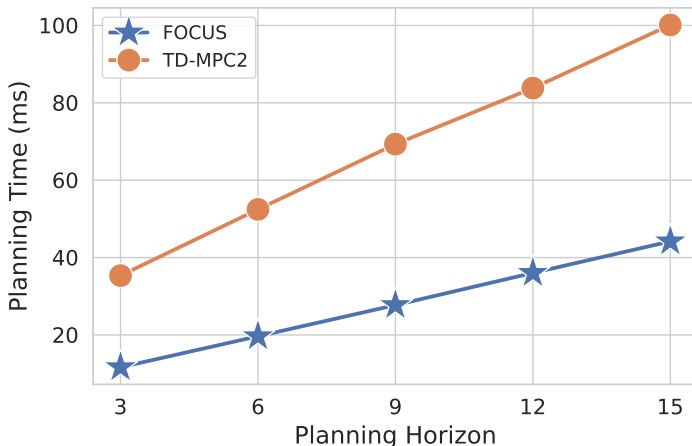

Figure 12: **Comparison of planning efficiency.** We measure the time per interaction step across different planning algorithms.

Table 2: **Episode return on `DMC` with low-dimensional action spaces.** The comparable performance to DreamerV3 highlights the applicability of FOCUS across diverse continuous visual control setups.

| Method | Walker Walk | Cheetah Run | Quadruped Walk | Average |
|---|---|---|---|---|
| FOCUS | 973.3±13.5 | 910.1±61.44 | 953.0±8.33 | 945.5 |
| DreamerV3 | 966.4±21.19 | 901.9±26.2 | 943.4±29.1 | 937.2 |
| TD-MPC2 | 332.9±31.9 | 202.7±2.3 | 162.2±49.7 | 232.6 |

## C.4 APPLICABILITY TO TASKS WITH LOW-DIMENSIONAL ACTION SPACES

A key concern regarding the proposed FOCUS is whether the high-level preference policy might hinder sufficient exploration in environments with small action spaces. To examine this, we evaluate FOCUS on four tasks from the DeepMind Control Suite characterized by low-dimensional action spaces: `Walker Walk`, `Cheetah Run`, and `Quadruped Walk`, while additional task details can be found in Table 1. As shown in Table 2, FOCUS achieves performance comparable to DreamerV3 across these low-dimensional control tasks, demonstrating its broad applicability.

## C.5 EXPERIMENTAL DETAILS ON HUMANOIDBENCH

**Setups.** The idea of diversity-driven hierarchical policy learning also extends to complex humanoid control tasks that use proprioceptive robot states as input. To validate this insight, we conduct additional experiments on two tasks from HumanoidBench (Sferrazza et al., 2024), namely `H1hand-push` and `H1hand-reach`.

**Model implementations.** We adopt the training framework of FastTD3 (Seo et al., 2025), which has been shown to efficiently solve HumanoidBench tasks through parallel simulation, large-batch updates, a distributional critic, and carefully tuned hyperparameters. We modify FastTD3's policy architecture to incorporate our hierarchical policy model and diversity-driven objectives, resulting in a variant that we refer to as **FastTD3-FOCUS**. Concretely, we implement the high-level preference policy as a 3-layer MLP that outputs Bernoulli logits, while keeping FastTD3's low-level policy architecture unchanged except for the ability to condition on preference prompts. During training, we compute both $\mathcal{L}_t^{\text{div}}$ and $\mathcal{L}_t^{\text{cst}}$ in addition to FastTD3's original policy loss. Since FastTD3 is a model-free algorithm, FastTD3-FOCUS follows the same setting and does not include world model learning or SMC planning.

**More explanations on the results.** Empirical results are presented in Figure 4 and Figure 16, comparing FastTD3 and FastTD3-FOCUS. This comparison helps illustrate the effect of our diversity-driven hierarchical policy learning objective, especially in tasks with relatively low-dimensional proprioceptive observations. While many HumanoidBench tasks (e.g., `H1hand-walk`,

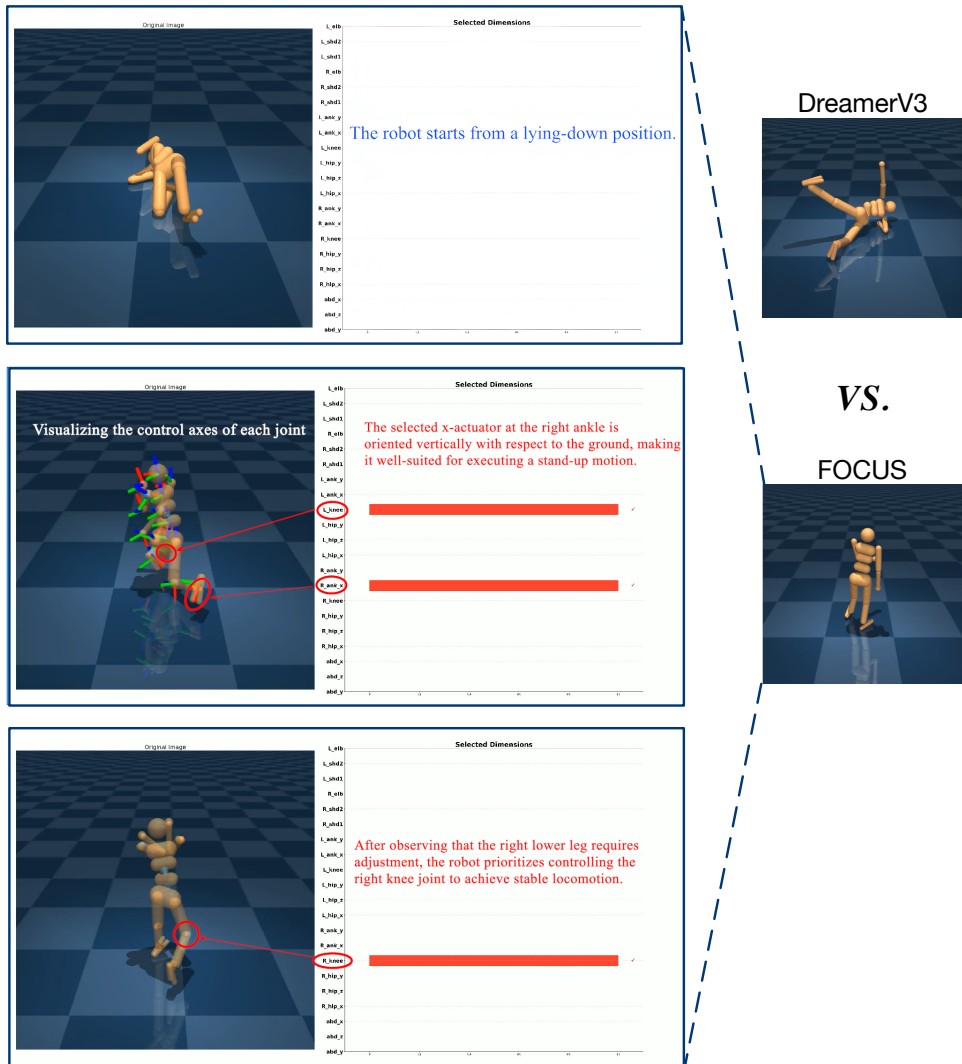

Figure 13: **Visualization of learned action-dimensional preferences.** Red bars indicate the action dimensions selected by the high-level policy. Please refer to our supplementary video for details.

`H1hand-stand`) become easy when trained using FastTD3's framework, several tasks remain challenging due to their high-dimensional action spaces. On both `H1hand-push` and `H1hand-reach`, FastTD3-FOCUS exhibits improved sample efficiency over the vanilla FastTD3 baseline, which uses a simple MLP policy. Notably, since the FastTD3-FOCUS results do not incorporate high-level SMC planning, the improvements provide clean evidence of the effectiveness of our diversity-driven learning objective on its own.

### C.6 HYPERPARAMETER ANALYSES ON DOGWALK

We further perform a hyperparameter sensitivity analysis on `DogWalk`, complementing the experiments on `HumanoidWalk`, with results shown in Figure 14(Left). As noted earlier, increasing $N_p$ does not necessarily improve performance, since the exclusive computation of the policy log-likelihood constrains the advantage of sampling more preference candidates.

### C.7 ABLATIONS ON CROSS-UNIT INTERDEPENDENCIES

We conduct ablation experiments in which dependencies across action dimensions are explicitly enforced, rather than sampled independently. Specifically, we apply a Softmax to the logits output by

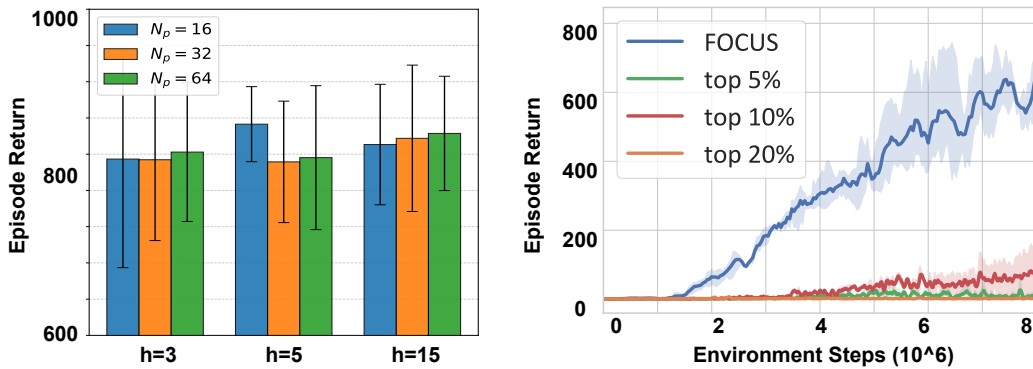

Figure 14: **Left:** Ablation study results on `DogWalk`. We evaluate the Effect of planning horizon $h$ and candidate trajectories $N_p$ in our planner. **Right:** Comparison of approaches that explicitly model inter-unit correlations. Test curves are averaged over three random seeds on `HumanoidWalk`.

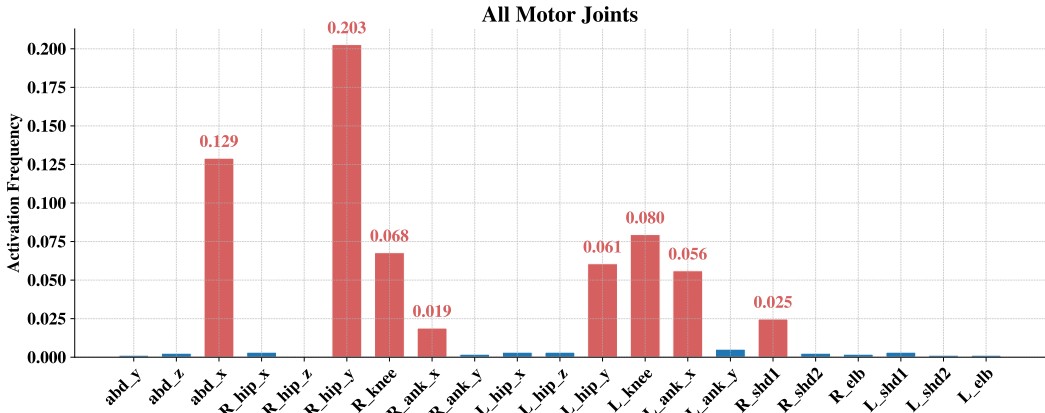

Figure 15: **Action-dimension selection frequencies.** FOCUS effectively leverages structured sparsity to guide focused exploration, supporting our objective of reducing exploration overhead in high-dimensional control tasks.

the high-level preference policy, followed by a Gumbel top-$k\%$ selection to determine the preference prompts. We test Top-5%, Top-10%, and Top-20% settings. The results, shown in Figure 14(Right), indicate that despite tuning the hyperparameter $k$, this design consistently fails to learn meaningful behaviors on the `HumanoidWalk` task. In contrast, our approach enables the low-level policy to implicitly capture correlations across control units, yielding more effective policy learning.

## C.8    FULL ACTION-DIMENSION SELECTION FREQUENCIES

We present the full action-dimension selection frequency in Figure 15, complementing Figure 7. Actuators beyond the top 8 most frequently selected ones are rarely activated, indicating that the high-level preference policy effectively discovers structural sparsity and reduces the action space.

## C.9    FULL RESULTS ON HUMANOIDBENCH

We present here the full empirical results on 6 additional tasks in HumanoidBench. The results in Figure 16 further demonstrate that FOCUS achieves significantly higher sample efficiency (e.g., `cabinet`, `room`) and the capability to avoid local optima (e.g., `truck`) in high-dimensional action spaces. For tasks like `crawl` and `slide`, our method maintains competitive performance comparable to the baseline.

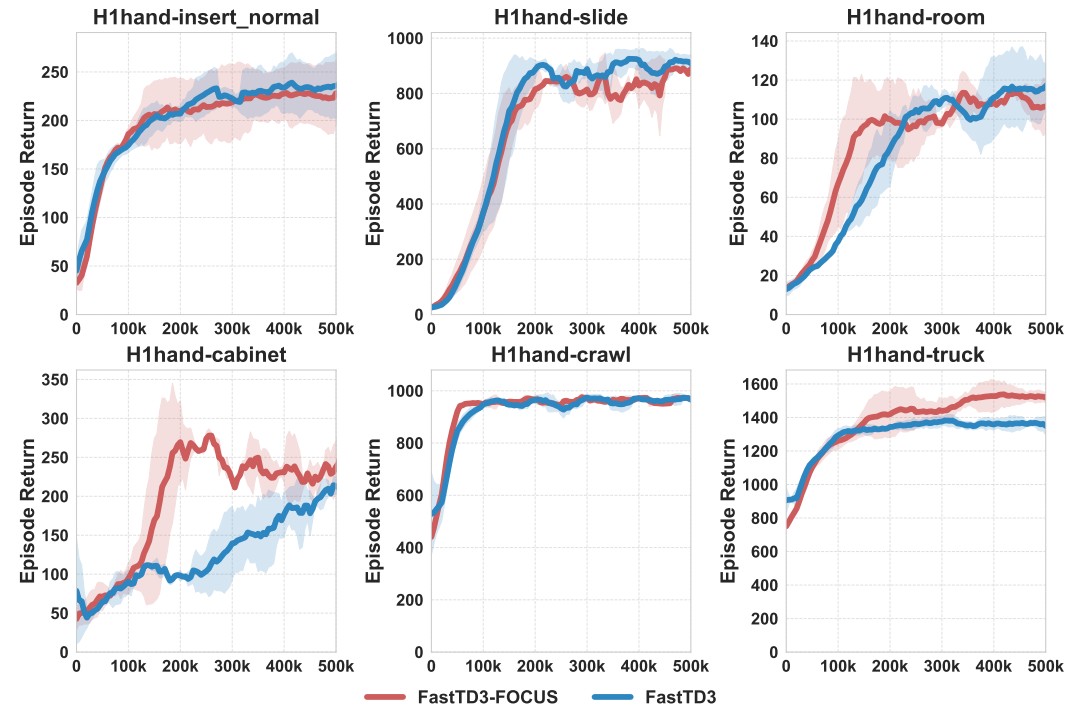

Figure 16: **Episodic returns of** 6 **tasks on HumanoidBench.**

## D USE OF LLMS

Large language models (LLMs) were used exclusively for language editing, including grammar correction and sentence polishing. They were not used for developing ideas, designing methods, conducting experiments, or interpreting results. Specifically, we provided prompts such as "Help polish this sentence to improve clarity and fluency in an academic writing style.", "Rewrite this sentence in a different expression", or "What is the alternative word for xxx". The output was then evaluated and further edited to stay close to this paper's writing style.

