# OpenReview forum: "FOCUS: High-Dimensional Model-Based RL via Focused Control Unit Sampling"
_ICLR.cc/2026/Conference — ICLR 2026 Conference Withdrawn Submission_

### Official Review · Reviewer_nvgW · 2025-10-27

**Soundness:** 3
**Presentation:** 3
**Contribution:** 3
**Rating:** 6
**Confidence:** 3

**Summary:**

This paper proposes a hierarchical approach for model-based reinforcement learning in which a high-level policy outputs binary prompts intended to focus exploration on certain action dimensions and a low-level policy outputs actions conditioned on these prompts.

**Strengths:**

- Intuitive approach solving a relevant problem.
- Strong performance across a suitable number of experiments and baselines.
- Well-written.

**Weaknesses:**

- The computational trade-off compared to "vanilla" DreamerV3 should be discussed more, especially in the context of B.4.
- The number of candidate trajectories is an important hyperparameter. The ablation appears to be based on a single run and the discussion is superficial.
- The explanation in Sec. 4.4 for why FOCUS w/o high-level performs worse than DreamerV3 is not convincing.
- Minor: Figure 6 should be enlarged

**Questions:**

- Is the presented observation wrt. the impact of the number of candidate trajectories robust and can you provide insights on why a higher number of samples would lead to worse performance?

---

> ### Author Response · Authors · 2025-11-24
> **Rebuttal by Authors-1**
>
> We appreciate the reviewer’s insightful comments. Please see our revised paper for the newly added results and corresponding visualizations.
>
> **Q1:** The computational trade-off compared to "vanilla" DreamerV3 should be discussed more.
>
> **A1:** We have conducted a detailed analysis of the total training time (including environment interaction, model learning, and planning) on an NVIDIA RTX 4090 GPU for the low-dimensional tasks in Appendix B.4 and the high-dimensional Humanoid tasks.
>
> The computational overhead in FOCUS primarily stems from two perspectives:
> - **Hierarchical Training:** Optimizing the additional high-level preference policy ($\pi_{\phi_1}$) and computing the diversity-driven auxiliary losses ($L_{div}, L_{cst}$).
> - **Inference-Time Planning:** The Preference-based Sequential Monte Carlo (SMC) planning performed during environment interaction.
>
> **Table R1: Wall-Clock Time Comparison between FOCUS and DreamerV3**
>
> | Domain | Env Steps | DreamerV3 (Baseline) | FOCUS (Ours) | Relative Increase |
> | :--- | :--- | :--- | :--- | :--- |
> | **Low-dim Tasks (Original B.4, now C.4)** *(Walker, Cheetah, etc.)* | 2M | ~12 hours | ~16 hours | +33% |
> | **High-dim Tasks**  *(Humanoid Domain)* | 8M | ~60 hours | ~72 hours | +20% |
>
> As shown in the table, compared with DreamerV3 in the low-dimensional tasks, FOCUS introduces approximately 33% computational overhead for the low-dimensional tasks. In this simple setting, DreamerV3 already solves the tasks efficiently (see Appendix B.4), meaning that the focused mechanism of our method yields comparable performance rather than a distinct advantage. This confirms that while FOCUS is generalizable to low-dimensional tasks, its computational cost makes it less efficient than "vanilla" DreamerV3 for simple problems where exploration is not the bottleneck.
>
> However, in the high-dimensional continuous control (e.g., Humanoid), the trade-off becomes highly favorable. The relative overhead decreases to 20% over a longer training horizon (8M steps). More importantly, this 20% increase in compute enables FOCUS to solve tasks that DreamerV3 struggles with (e.g., Humanoid Walk in Figure 2), or to achieve higher sample efficiency.
>
> **Q2:** The number of candidate trajectories is an important hyperparameter. The ablation appears to be based on a single run and the discussion is superficial.
>
> **A2:** We have extensively examined hyperparameter tuning in the revised paper and results are presented in **Fig.5(b) and Fig.14**. We also included an in-depth analysis in Section4.5. Below we present the numeric results on both Humanoid Walk and Dog Walk respectively in Table1 and Table2.
>
> **Table R1: Humanoid Walk Performance with different hyperparameters**
>
> |      | $N_p$=16        | $N_p$=32        | $N_p$=64        |
> |:---- |:--------------- |:--------------- |:--------------- |
> | h=3  | 788.45 ± 161.86 | 806.05 ± 127.68 | 807.41 ± 125.86 |
> | h=5  | 804.77 ± 124.37 | 809.96 ± 109.85 | 843.09 ± 103.61 |
> | h=15 | 839.12 ± 183.20 | 815.94 ± 116.47 | 833.57 ± 125.66 |
>
> **Table R2: Dog Walk Performance with different hyperparameters**
>
> |      | $N_p$=16        | $N_p$=32        | $N_p$=64       |
> |:---- |:--------------- |:--------------- |:-------------- |
> | h=3  | 843.27 ± 149.80 | 842.31 ± 111.54 | 852.90 ± 95.80 |
> | h=5  | 891.40 ± 51.79  | 839.38 ± 83.80  | 845.21 ± 99.30 |
> | h=15 | 863.29 ± 83.15  | 871.90 ± 101.00 | 878.66 ± 78.87 |
>
> We notice that the discussion to these results directly address the concern raised in Q4. To avoid redundancy, we have dedicated the response to Q4 to a comprehensive discussion on why higher sample counts does not necessarily bring performance upgrade.

---

> > ### Author Response · Authors · 2025-11-24
> > **Rebuttal by Authors-2**
> >
> > **Q3:** The explanation in Sec. 4.4 for why FOCUS w/o high-level performs worse than DreamerV3 is not convincing.
> >
> > **A3:** The explanation in Section 4.4 regarding the "w/o high-level" variant was not sufficiently clear. We would like to clarify the exact configuration of this ablation and provide additional experimental evidence to support our claims.
> >
> > 1. **Exact setting of the "w/o $\pi_{\phi_1}$" Variant:**
> >     The "w/o high-level" variant is not simply a standard DreamerV3 plus planning model. Instead, it is a DreamerV3 agent that employs Sequential Monte Carlo (SMC) planning in the flat, high-dimensional action space during **both the exploration (train buffer collection) and policy evaluation**. The poor performance of this variant compared to DreamerV3 stems from this **biased exploration**, especially when value function is noised during early stages of training. This degrades the replay buffer quality and hampers the convergence of the world model and policy, resulting in lower performance than the standard DreamerV3.
> >
> > 2. **DreamerV3 with Inference-Only Planning:**
> >     To isolate the impact of planning in high-dimensional spaces, we conducted an additional experiment on "DreamerV3 w. planning". In this setup, we applied the SMC planner to DreamerV3 during evaluation only. The ablation results are extended in Table3:
> >
> >     **Table R3: Extensive Ablation Study Results**
> >
> >     | Ablates               | Mean | Std  |
> >     |:--------------------- |:---- |:---- |
> >     | FOCUS                 | 6030 | 1856 |
> >     | w/o high-level        | 2354 | 2363 |
> >     | w/o diversity         | 4747 | 2860 |
> >     | w/o planning          | 4690 | 1889 |
> >     | Dreamerv3             | 3281 | 2346 |
> >     | Dreamerv3 w. planning | 5249 | 4465 |
> >
> >     While adding planning to DreamerV3 during evaluation phase improves the mean score, it introduces massive instability evidenced by the extremely high standard deviation. This indicates that planning in the flat high-dimensional space is unstable, where the planner is vulnerable to getting stuck in local optima. In contrast, FOCUS achieves the best mean(6030) with the smallest standard deviation(1856), demonstrating that high-level planning with promotion for diversity also effectively bypasses the optimization traps.
> >
> > We have included the results of "DreamerV3 w. planning" variant in Fig. 5(a), together with a clearer discussion in Section 4.5 in the revised paper.
> >
> > **Q4:** Is the presented observation wrt. the impact of the number of candidate trajectories robust and can you provide insights on why a higher number of samples would lead to worse performance?
> >
> > **A4:** We have revised our paper to include the discussion on hyperparameter analysis results shown in Fig. 5 in Section 4.5. While the standard SMC planning suggests monotonic (though often marginal) improvement with the increasement of $N_p$, our results in the above Table R1 and Table R2 suggest otherwise. These results can be attributed to the consequence of **the exclusive use of the policy log-likelihood in the advantage calculation**. Our practise of removing log-likelihood prevents the massive variance it introduces, but also encourages riskier behaviour for the planner. As the number of trajectories ($N_p$) increases, the planner becomes increasingly dominated by high-value but low-probability action trajectories, ultimately degrading overall planning performance rather than improving it.

---

### Official Review · Reviewer_7vBd · 2025-10-28

**Soundness:** 2
**Presentation:** 2
**Contribution:** 2
**Rating:** 4
**Confidence:** 4

**Summary:**

This paper proposes FOCUS, a hierarchical model-based RL framework for visual continuous-control tasks with high-dimensional action spaces, where a high-level “preference policy” samples binary control-unit prompts (one per action dimension) and a low-level policy conditions on these prompts to focus exploration and execution on salient dimensions. Empirically, FOCUS outperforms strong baselines (DreamerV3, TD-MPC2, DrQv2, Director) on DMC Humanoid/Dog and MyoSuite hard tasks under matched interaction budgets, with ablations showing all components contribute. Main questions are robustness of the binary prompt design (vs. softer/structured sparsity), the need for predefined action factorization, and fuller reporting of compute costs; nonetheless, the approach is a practical step toward scalable MBRL in very high-dimensional actions.

**Strengths:**

1. The idea is well-motivated—only a sparse subset of joints often matters at a time—and the mechanism is clearly specified, including the Bernoulli-coded prompts and planner weighting/update rules.

**Weaknesses:**

1. Insufficient benchmarking. The authors only picked 4 tasks and didn't include Humanoidbench, which, to my knowledge, could be the most suitable benchmark for this paper.

2. Inappropriate setting. In my understanding, the most important contribution of this paper is identifying the significant action dimensions. It is not necessary to bind this paper to visual input, or at least, authors should consider the proprio inputs as one of the experimental settings.

**Questions:**

See above,

---

> ### Author Response · Authors · 2025-11-24
> **Rebuttal by Authors**
>
> **Q1 & Q2:** Insufficient benchmarking & Inappropriate setting.
>
> We greatly appreciate the reviewer’s suggestion to include **HumanoidBench** and to validate our method on **proprioceptive inputs**. Below, we address both points with new experimental evidence on HumanoidBench using low-dimensional proprioceptive observations.
>
> In Section 4.4 in the revised paper, we have added experiments on two challenging HumanoidBench tasks, namely **H1hand-push** and **H1hand-reach**. Concretely, we extend the **FastTD3** framework [1] framework, which is known for solving HumanoidBench tasks with high computational efficiency. We incorporate our hierarchical policy design and diversity-driven objectives ($\mathcal{L}_t^{\text{div}}$ and $\mathcal{L}_t^{\text{cst}}$) into FastTD3, creating a variant we call **FastTD3-FOCUS**.
>
> The comparison results between FastTD3 and FastTD3-FOCUS is presented in **Fig. 4**. As shown, FastTD3-FOCUS consistently obtains **improved performance** on H1hand-push. On H1hand-reach, while both methods reach convergence with similar episodic returns, FastTD3-FOCUS shows **higher sample efficiency**. We attribute this to its faster coverage of high-return regions in the state space, which accelerates the acquisition of effective reaching behaviors. These results demonstrate the applicability of our method to HumanoidBench under proprioceptive inputs.
>
> [1] Seo, Younggyo, et al. "FastTD3: Simple, Fast, and Capable Reinforcement Learning for Humanoid Control." arXiv preprint arXiv:2505.22642 (2025).

---

> > ### Comment · Reviewer_7vBd · 2025-11-24
> > **Insufficient benchmarking on HumanoidBench**
> >
> > Thanks for authors' responses and updates. It would be better if authors can include more humanoidbench tasks. Two tasks are insufficient to demonstrate the efficacy.

---

> ### Author Response · Authors · 2025-11-29
> **Rebuttal by Authors**
>
> **Q:** Two tasks are insufficient to demonstrate the efficacy.
>
> We thank the reviewer for the prompt feedback and the suggestion to expand the evaluation on HumanoidBench. We agree that demonstrating efficacy across a broader range of tasks strengthens the validation of our method. In response , we have conducted **extensive experiments on 6 additional tasks from HumanoidBench**, covering a diverse set of challenges including manipulation and locomotion (specifically: `H1hand-cabinet`, `H1hand-crawl`, `H1hand-insert_normal`, `H1hand-room`, `H1hand-slide`, and `H1hand-truck`).
>
> We have revised our paper to present the results in **Fig.16**. As illustrated in the figure, FastTD3-FOCUS (Our modified variant) achieves higher sample efficiency compared to FastTD3 in tasks such as `cabinet`, `crawl`, `room` and `slide` given that they converge to similar performance. While in `insert_normal` and `truck`, FastTD3-FOCUS achieves superior performance which demonstrates its capability to avoid suboptimal local optima during policy learning in high-dimensional action space. Note that due to limited time window, the curve now consists of only one seed and we are currently running experiments with additional seeds. We will update a revised version as long as all experiments are finished.

---

> > ### Author Response · Authors · 2025-12-04
> > **Follow-up: Full multi-seed results for the 6 additional HumanoidBench tasks**
> >
> > Dear Reviewer 7vBd,
> >
> > Following up on our previous response, we have now completed the experiments with $3$ seeds for all the $6$ additional HumanoidBench tasks. We have updated a revised paper with updated Figure.16 which showcases both the mean and standard variance. As illustrated, the full results consistently validate the effectiveness of FastTD3-FOCUS:
> >
> > 1. Higher Sample Efficiency & Performance: Our method demonstrates significantly faster convergence and higher asymptotic performance on complex manipulation tasks such as `cabinet` and `room`.
> > 2. Escaping Local Optima: On `truck`, the baseline plateaus at a sub-optimal level, whereas FastTD3-FOCUS successfully learns a better policy, highlighting its ability to explore high-dimensional action spaces effectively.
> > 3. Robustness: On locomotion tasks like `crawl`, our method maintains competitive performance comparable to the strong baseline.
> >
> > We believe these extensive results, covering both manipulation and locomotion, strongly support the efficacy of our method across diverse scenarios. We hope this fully addresses your concern regarding the benchmarking sufficiency.
> >
> > Best regards,
> >
> > The Authors

---

### Official Review · Reviewer_MvbX · 2025-10-30

**Soundness:** 3
**Presentation:** 2
**Contribution:** 3
**Rating:** 4
**Confidence:** 3

**Summary:**

FOCUS is a hierarchical model-based RL method for high-dimensional continuous control. A high-level “preference” policy samples a binary prompt over control units, and a low-level policy outputs actions conditioned on that prompt. During Dreamer-style imagination, diversity and consistency regularizers make selected units behave differently and unselected ones similarly, so prompts actually influence actions. After that, the agent does preference-based SMC planning: it samples several prompts, rolls out short imagined futures with each fixed prompt, weights them by exponentiated advantages, and executes the first action, yielding stronger performance and lower decision latency than baselines on benchmarks.

**Strengths:**

1. The paper is generally well written.
2. The proposed method is novel and shows promising results compared to the baselines.

**Weaknesses:**

1. Prompts are sampled from independent Bernoulli variables, yet the paper claims to capture inter-unit correlations; the modeling and empirical effect of such dependencies are not clearly demonstrated.
2. The claim that prior knowledge about important dimensions can be incorporated is not evaluated with targeted experiments.

**Questions:**

1. How exactly are “control units” defined per task?
2. Could you explain the motivation for exponentiating the advantage when updating particle weights? This choice can increase variance. How do you address stability?

---

> ### Author Response · Authors · 2025-11-24
> **Rebuttal by Authors-1**
>
> We appreciate the reviewer’s insightful comments. Please refer to our revised paper for the newly added results and corresponding visualizations.
>
> **Q1:** Prompts are sampled from independent Bernoulli variables, yet the paper claims to capture inter-unit correlations; the modeling and empirical effect of such dependencies are not clearly demonstrated.
>
> **A1:** Despite the independent sampling process, co-optimizing the high-level and low-level policies implicitly facilitates the discovery of interdependencies among action dimensions. To illustrate this, in Appendix C.7, we conduct an additional experiment where inter-unit dependencies are explicitly enforced during sampling. Specifically, instead of Bernoulli sampling, we apply a Softmax to the logits output by high-level preference policy, followed by a **Gumbel Top k%** selection. We tested Top-5%, Top-10%, and Top-20% settings.
>
> **Q2:** The claim that prior knowledge about important dimensions can be incorporated is not evaluated.
>
>
> **A2:** We acknowledge that while the hierarchical design in FOCUS may enable more convenient incorporation of prior knowledge, our main focus in this work is on the automatic discovery of important action dimensions, and we have not conducted corresponding experiments. Therefore, we have removed the related description in the updated paper to avoid overclaiming our contributions.
>
> Nonetheless, this remains a valuable direction for future work. Specifically, the incorporation of prior knowledge can be achieved via a masking scheme: if a set of important actuators is known as priori, a hard mask $\mathbf{m}_t$ can be applied to the preference prompt, forcing the policy to focus solely on the specified dimensions. The output of high-level policy can be foumulated as $\mathbf{m}_t \odot \mathbf{u}_t$.
> As shown in **Fig. 14(right)**, the Top-K variants fail to learn meaningful behaviours on the Humanoid Walk task, suggesting optimization difficulties introduced by explicit interdependency modeling. In contrast, our approach allows the low-level policy to implicitly grasp correlations across control units, resulting in more effective policy learning. These results are included in the appendix, and the methodology section has been updated accordingly to clarify our design choice.
>
> **Q3:** How exactly are “control units” defined per task?
>
> **A3:** Each control unit is defined to correspond to a subset of action dimensions for a specific functional component. Taking Humanoid Walk as an example, each action dimension represents an actuator functioning along specific axis(e.g. X, Y and Z), while a control unit can span multiple actuators to model inter-actuator and even inter-joint dependencies.

---

> ### Author Response · Authors · 2025-11-24
> **Rebuttal by Authors-2**
>
> **Q4:** Could you explain the motivation for exponentiating the advantage when updating particle weights? This choice can increase variance. How do you address stability?
>
> **A4:** It is a very reasonable concern! Our choice of exponentiating the advantage for the importance weights follows the *Control as Inference* framework as detailed in [1].
>
> (1) *Why exponentiating the advantage?*
>
> Intuitively, since our goal is to approximate the optimal posterior using samples from proposal policy, exponentiating the advantage casts a stronger **selective impact on those particles with higher advantages**. This allows the algorithm to quickly distinguish and focus on high-quality particles, which is crucial for efficient planning in limited horizons.
>
> We fisrt present experimental results on removing the exponential on advantages when updating particle weights in the table below. As can be seen, this ablation hurts the performance under both short and long horizons, indicating the dominant role exponentiation plays in distinguish high-quality particles.
>
> **Table R1: Episode Return on Humanoid Walk without exponential weight update**
>
> | Setting          | h=3                 | h=15            |
> |:---------------- |:------------------- |:--------------- |
> | w/o. exponential | 760.39 ± 100.77     | 775.47 ± 143.57 |
> | w. exponential   | **835.27 ± 133.42** | 807.79 ± 127.85 |
>
> (2) *How to alleviate high variance?*
>
> The variance is a major concern, especially when the horizon $T$ is long. A common approach to limiting variance is **particle resampling** [2], where particles are stochastically resampled toward higher-likelihood regions. This procedure effectively reduces the growth of estimation variance from exponential in $t$ to linear. Concretely, after updating particle weights, trajectories are resampled proportionally to their weights to form a new particle set, and weights are reset to uniform thereafter.
>
> We further observe that during online planning, **removing the log-likelihood term** in the advantage computation significantly reduces variance and improves performance. We argue that while removing this term brings a biased estimation and risky estimations towards actions with high advantage and low likelihood, it also gets rid of the massive variance introduced in high-dimensional continuous action spaces. This trade-off results in more stable planning and mitigates particle degeneration, ultimately leading to stronger empirical performance.
>
> To validate these observations, we compute the harmonic **Effective Sample Size** (the closer to 100% the better) as in [2] to show the impact of particle resampling and the impact of removing log-likelihood. As shown, removing the log-likelihood plays a critical role in reducing variance when dealing with high-dimensional continuous action spaces, while particle weight resampling also helps, especially when the planning horizon is long.
>
> **Table R2: Harmonic ESS on Humanoid Walk (without logpi)**
>
> | Setting         | h=3        | h=15          |
> |:--------------- |:---------- |:------------- |
> | w. Resampling   | 94.4% ± 3% | 95.3% ± 3.9%  |
> | w/o. Resampling | 91.9% ± 5% | 87.6% ± 10.7% |
>
> **Table R3: Harmonic ESS on Humanoid Walk (with logpi)**
>
> | Setting         | h=3          | h=15         |
> |:--------------- |:------------ |:------------ |
> | w. Resampling   | 3.7% ± 0.09% | 3.5% ± 0.03% |
> | w/o. Resampling | 3.1% ± 0.06% | 2.2% ± 0.06% |
>
> **References:**
> [1] Reinforcement Learning and Control as Probabilistic Inference: Tutorial and Review
>
> [2] Probabilistic Planning with Sequential Monte Carlo methods

---

### Official Review · Reviewer_2GBj · 2025-11-01

**Soundness:** 2
**Presentation:** 2
**Contribution:** 2
**Rating:** 4
**Confidence:** 3

**Summary:**

The paper presents FOCUS, a model-based reinforcement learning (MBRL) framework for high-dimensional continuous control that enhances efficiency by focusing on key action dimensions. It employs a hierarchical policy, where a high-level module samples binary prompts to select important control units, and a low-level policy generates conditioned actions. A diversity-driven objective promotes exploration, and a preference-based Monte Carlo planner improves decision quality. Experiments on DeepMind Control Suite and MyoSuite show FOCUS outperforming baselines like DreamerV3 and TD-MPC2 in sample efficiency and performance.

**Strengths:**

1. Well-written and clearly presented, concepts like preference prompts, interaction policy, and focused sampling are explained intuitively and supported by clear figures.

2. Target a core challenge in reinforcement learning: scaling model-based methods to high-dimensional action spaces.

**Weaknesses:**

1. It lacks a thorough discussion of prior studies that also tackle high-dimensional control challenges ([1], [2], [3]). A more comprehensive literature review would provide clearer connections to existing research and better highlight the paper’s contributions.

2. The two-level policy design, combined with diversity regularization and SMC-based planning, introduces additional structural and computational complexity compared to single-policy MBRL approaches. The trade-off between performance gains and computational cost is not clearly analyzed or quantified.

3. Although the framework’s effectiveness is demonstrated empirically,  it lacks formal theoretical justification regarding how preference prompts influence optimization dynamics, stability, or convergence.

[1]  Schumacher, Pierre, et al. "DEP-RL: Embodied Exploration for Reinforcement Learning in Overactuated and Musculoskeletal Systems." The Eleventh International Conference on Learning Representations.

[2] Zhong, Dianyu, Yiqin Yang, and Qianchuan Zhao. "No prior mask: Eliminate redundant action for deep reinforcement learning." Proceedings of the AAAI Conference on Artificial Intelligence. Vol. 38. No. 15. 2024.

[3] Zhang, Wenbo, and Hengrui Cai. "Where to Intervene: Action Selection in Deep Reinforcement Learning." Transactions on machine learning research (2025).

**Questions:**

1. Could the authors provide more fine-grained statistics or visualizations of the selection frequency across action dimensions over time? This would help illustrate how the preference policy dynamically shifts focus among control units.

2. Could the authors elaborate on the time cost associated with world model learning versus environment interaction? A breakdown of computational cost would clarify where the main bottlenecks occur.

---

> ### Author Response · Authors · 2025-11-24
> **Rebuttal by Authors-1**
>
> We thank the reviewer for the valuable comments. Please refer to our revised paper for the newly added results and corresponding visualizations.
>
> **Q1:** It lacks a thorough discussion of prior studies that also tackle high-dimensional control challenges.
>
> **A1:** We have updated our paper to include these related works in Section 5. Below, we provide a detailed comparison:
> - **DEP-RL:** DEP-RL introduces a noise correlated across actions to deal with musculoskeletal overactuated systems, which **requires prior knowledge** on the mapping between muscle length and control signal. FOCUS, on the other hand, acts without any prior knowledge. Moreover, DEP exploration is integrated in RL as a gradient-free data provider, while FOCUS optimizes to identify prioritized control units through a preference policy.
> - **No Prior Mask:** This method learns action masks that filter out actions having a minor impact on state transitions, which resembles our intuition in reducing the exploration space. However, the masking-out strategy is specific to tasks with **discrete action spaces** and is hard to apply to continuous tasks, where simply masking out some action dimensions leads to task failures. FOCUS learns to attribute more exploration to relevant action dimensions and suppress that on less relevant ones, achieving effective policy optimization in high-dimensional continuous action spaces.
> - **Action Selection:** This approach resembles our method in identifying important action dimensions without using prior knowledge in high-dimensional action spaces. To incorporate the action selection results into RL, it forces an **unchanged hard action mask** to remove action dimensions from the identified minimal sufficient action set. However, FOCUS selects action dimensions by prioritized control units, which vary over time through a high-level preference policy, in a soft manner to encourage and suppress exploration accordingly.
>
> In summary, although these methods share the motivation of reducing action space to improve learning efficiency, they fundamentally differ from FOCUS in **generality** and **adaptability**. DEP-RL relies on domain-specific priors, No Prior Mask is limited to discrete action spaces, and Action Selection enforces static hard constraints. By contrast, FOCUS learns a dynamic, soft preference policy that prioritizes control units in high-dimensional continuous spaces without any prior physical knowledge.
>
>
> **Q2:** The trade-off between performance gains and computational cost is not clearly analyzed or quantified.
>
> **A2:** Given the significant performance gains achieved by the two-level policy in FOCUS, the additional structural and computational complexity is negligible.
> - **Structural Complexity:** The proposed hierarchical policy model only extends one 3-layer MLP to two, resulting in a minimal increase in the overall network size.
> - **Computational Complexity:** The primary concern might lie in calculating the diversity-driven objective in the SMC-based planning. However, the complexity can be reduced by parallelizing the summation over $N_c$ and $d$. As shown in Figure 10, the SMC-based planning is more efficient compared to MPPI in TD-MPC2.
>
> Notably, the two-level policy effectively reduces the high-dimensional action space and enables a more focused exploration within compact control units. This achieves higher learning efficiency, which compensates for the additional computational complexity.
>
> For a clearer perspective on the trade-off between performance gains and computational overhead, we summarize the runtime and task performance of FOCUS, TD-MPC2 and vanilla DreamerV3 below on the DMC Humanoid Walk task. All experiments are conducted on a single RTX 4090 with a fixed environment step budget of 8M.
> - FOCUS: About $72$ hours of training and an average of $709$ episodic return.
> - TD-MPC2: About $221$ hours of training and an average of $2$ episodic return.
> - DreamerV3: About $70$ hours of training and an average of $82$ episodic return.
>
> As demonstrated, FOCUS shows both sample efficiency and computational efficiency compared to TD-MPC2 on high-dimensional continuous control tasks with pixel inputs. While introducing minor structrual and computational complexity to vanilla DreamerV3, it brings about considerable performance improvement.

---

> ### Author Response · Authors · 2025-11-24
> **Rebuttal by Authors-2**
>
> **Q3:** The paper lacks formal theoretical justification regarding how preference prompts influence optimization dynamics, stability, or convergence.
>
> **A3:** We admit that theoretical justification regarding how preference prompts influence optimization dynamics, stability, or convergence is challenging. To compensate, we have provided thorough empirical results in the revised paper: (i) In **Fig. 7**, the visualization of most frequently selected action dimensions indicates a successful discovery of meaningful structral sparsity; (ii) In **Fig. 8**, we also include a temporal evolution heatmap to show the dynamic selection strategy. We hope these results would provide intuitive insights into the optimization dynamics and justify the effectiveness of our proposed mechanism.
>
>  **Q4:** Could the authors provide more fine-grained statistics or visualizations of the selection frequency across action dimensions over time? This would help illustrate how the preference policy dynamically shifts focus among control units.
>
> **A4:** Sure! We have included Fig. 8 in the revised paper, a heatmap to show how the preference logits change over 100 steps within an episode of Humanoid Walk, aiming at facilitating a more comprehensive understanding of the preference policy behavior. The high-level preference policy employs a sparse and dynamic selection strategy, with specific action dimensions being selectively activated in response to temporal transitions.
>
> **Q5:** Could the authors elaborate on the time cost associated with world model learning versus environment interaction? A breakdown of computational cost would clarify where the main bottlenecks occur.
>
> **A5:** To clarify the time cost associated with world model learning versus environment interaction, we measured the average wall-clock time per step during the training phase. The breakdown is as follows:
>
> - **Environment Interaction**: 0.37s per step
> - **Policy Inference**: 0.48s per step
> - **Agent Training**: 0.68s per step
>
> The **Agent Training** includes the world model learning and policy learning. The results indicate that the training process is the primary computational bottleneck, taking approximately 1.8 times longer than environment interaction. The policy inference takes about 1.2 times longer than environment interaction due to the online planning.
>
> However, we would like to highlight that while the per-step computational cost is higher than DreamerV3, FOCUS improves sample efficiency considerably and thereby reduces the total number of interaction steps required to solve the task, as shown in the DMC (Fig. 2), Myosuite (Fig. 3), and the newly added HumanoidBench (Fig. 4) results.

---

### Author Response · Authors · 2025-11-24
**General Response**

**We are grateful to the reviewers for their insightful comments and valuable suggestions. In response, we have updated our paper and a list of major changes is presented below.**

1. **Incorporated a new experimental domain and a new baseline model (Section 4.4, Appendix C.5 & Appendix C.9):** We further evaluated our approach on HumanoidBench, a challenging high-dimensional control benchmark, and compared our approach against a strong baseline, FastTD3. Results for two tasks are illustrated in Figure 4, with the additional six tasks presented in Figure 16 in Appendix C.9.
2. **Added an ablation study and discussion on cross-action-unit interdependencies (Section 3.2 & Fig. 14):** We clarifed how our method handles correlations in Section 3.2 and added a new experiment to compare alternative approaches for capturing action interdependencies in Figure 14(right) in Appendix C.7.
3. **Included a more thorough hyperparameter analyses (Section 4.5 & Appendix C.6):** We provided standard variance across 20 runs to analyze the effect of planning horizon ($h$) and the number of candidate trajectories ($N_p$) on both HumanoidWalk and DogWalk tasks, along with a deeper discussion on the hyperparameter influences.
4. **Provided visualization of preference logits over time (Section 4.6):** We added Fig. 8 to visualize the evolution of action preference logits within a $100$ timestep window, showing that our high-level preference policy employs a sparse and dynamic selection strategy.
5. **Expanded the Related Work section (Section 5):** We incorporated a broader discussion on high-dimensional control and action redundancy, citing more relevant studies (Schumacher et al., 2022; Zhong et al., 2024; Zhang & Cai, 2025).
6. **Restructured the Appendix for better clarity:** We added new results, including investigations of alternative high-level policy network designs that enforce explicit dependencies across action dimensions, rather than sampling them independently (Appendix C.7).

We hope that these revisions resolve the reviewers' concerns, and are more than willing to address any further comments from them.

Sincerely,

The Authors

---

### Author Response · Authors · 2025-12-04
**Rebuttal Summary for AC**

1. **Summarization of Technical Novelty in FOCUS**

   FOCUS introduces a new model-based reinforcement learning framework that learns action-dimensional preferences to tackle exploration inefficiency in high-dimensional action spaces. Unlike prior work that treats all action dimensions equally, FOCUS (i) infers low-dimensional preference structures revealing which action subspaces meaningfully drive system dynamics, (ii) adapts model-based rollouts based on these learned preferences to yield more targeted policy optimization, and (iii) integrates preference guidance to perform high-level planning which helps avoiding local optima. This results in improved sample efficiency, stability, and final performance across diverse continuous-control tasks.

2. **Main Concerns Raised by Reviewers**

   - Insufficient Benchmarking: Reviewers requested validation on more challenging high-dimensional benchmarks (specifically HumanoidBench) and settings using proprioceptive inputs to prove robustness beyond pixel inputs.
   - Computational Trade-offs: Concerns were raised regarding the computational efficiency compared to baselines and whether the performance gains justify the cost.
   - Methodological Clarifications: Questions arose regarding specific design choices.
   - Literature: Reviewers noted a lack of discussion on specific high-dimensional control works.

3. **Our Response and Major Modifications in the Revision**

   **Expanded Benchmarking (HumanoidBench)**: We extended our evaluation to HumanoidBench with proprioceptive inputs. We implemented FastTD3-FOCUS, demonstrating that our hierarchical mechanism consistently improves sample efficiency and performance over the FastTD3 baseline on complex manipulation and locomotion tasks.

   **Comprehensive Ablation & Analysis**: We added extensive studies to address methodological concerns:

   - Correlations: We compared our implicit learning against explicit dependency modeling (Top-K sampling), showing our design choice is superior (Fig. 14 Right).
   - Planning Stability: We validated the design choices of exponentiating advantages and removing log-likelihood terms to manage variance in high-dimensional spaces.
   - Hyperparameters: We provided a detailed analysis of particle counts ($N_p$) and planning horizons ($h$).
   - Computational Cost Analysis: We provided a breakdown of training time consumed.
   - Ablation clarity: We added a "DreamerV3 with inference-only planning" ablation in Fig.5(a) to differentiate from "FOCUS w/o. $\pi_{\phi_1}$".
   - Preference shift: We added heatmaps (Fig. 8) visualizing the dynamic shift of preference logits over time.

   **Literature**: We expanded Section 5 to distinguish FOCUS from prior static masking or physics-prior-based methods.

4. **Post-Rebuttal Discussion Summary**

   Following the rebuttal phase, reviewer feedback evolved in two main ways:

   - Positive updates: Reviewer nvgW confirmed that the concerns regarding computational trade-offs and ablations were fully addressed, leading to a recommendation for a clear accept.
   - Request for experimental expansion: Reviewer 7vBd initially noted that the two HumanoidBench tasks added in the initial rebuttal were insufficient to fully demonstrate efficacy. In response, we immediately conducted experiments on 6 additional HumanoidBench tasks. These results further validated that FOCUS achieves higher sample efficiency while also avoids suboptimal local optima where the baseline fail.

5. **Final Remarks**

   We appreciate the reviewers’ and AC’s careful assessment. We believe the revised paper now presents a clear and compelling technical contribution, thoroughly supported by the inclusion of the challenging HumanoidBench suite and deep ablation studies. We hope this strengthened version resolves all concerns and demonstrates the significance of FOCUS.

---

### Note · Authors · 2026-01-19

**Comment:**

Dear Area Chairs and Reviewers,

After careful consideration of the reviews and feedback, we have decided to withdraw this submission. We sincerely appreciate the time and effort the reviewers dedicated to evaluating our work.

Best Regards,

Authors

**Withdrawal Confirmation:**

I have read and agree with the venue's withdrawal policy on behalf of myself and my co-authors.